# Breaking the Synthetic-Real Domain Shortcut for Training-Free Generative Replay-based Class Incremental Learning

Tao Zhang [1 2]   Qixuan Fan [1]   Yiyuan Liang [1 2]   Yanjie Wang [1 2]   Song Yan [3]   Tian Tian [1 2]   Jiahuan Zhou [4]
Luxin Yan [1 2]   Sheng Zhong [1 2]   Xu Zou [* 1 2]

## Abstract

Class-incremental learning (CIL) requires models to continuously acquire new knowledge while avoiding catastrophic forgetting. While exemplar replay is effective, it raises concerns regarding privacy and storage. Thus, generative replay has emerged as a viable alternative, synthesizing old data using frozen pretrained text-to-image (T2I) models without any extra training. However, we observe that directly mixing synthetic old-class data with real new-class data during incremental training leads to significant performance degradation. This issue stems from a "domain shortcut", where models rely on domain-discriminative features instead of semantic class cues. To address this, we propose DREAM (**D**omain-**R**egularized **E**xemplar-free **A**lignment **M**odel), which uses a training-free generator to synthesize old-class data and eliminates domain shortcut via subspace rectification and orthogonal projection, while reinforcing semantic alignment through real-anchored prototype regularization. Extensive experiments on 4 datasets demonstrate that DREAM outperforms existing exemplar-free CIL methods and achieves state-of-the-art performance. Our source code is available at https://github.com/Light-ZhangTao/DREAM.

## 1. Introduction

Real-world applications demand models capable of Continual Learning (CL) to handle evolving data streams, yet deep networks suffer from catastrophic forgetting (McCloskey

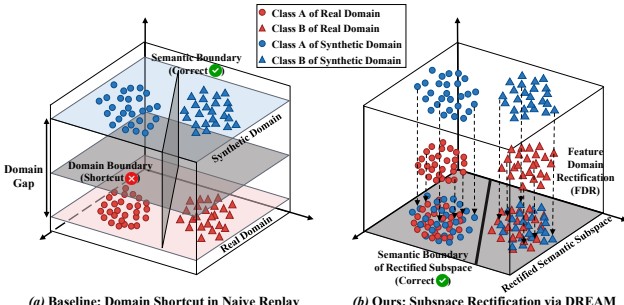

*Figure 1.* Illustration of the "Domain Shortcut" phenomenon and Our Solution. **(a) Baseline:** In standard generative replay, a significant domain gap between synthetic old-class data (blue) and real new-class data (red) creates a "Domain Shortcut". This gap causes the model to mistakenly treat the real space as the new class space, prioritizing domain separation over semantic class distinction during inference. **(b) DREAM (Ours):** We propose Feature Domain Rectification (FDR) to excise "Domain Shortcut" via Orthogonal Subspace Projection. By projecting features into a rectified semantic subspace, the domain gap is eliminated, forcing the model to learn the true semantic decision boundary (black line) and effectively aligning real and synthetic distributions.

& Cohen, 1989; Goodfellow et al., 2013) when trained sequentially. Consequently, Class-Incremental Learning (CIL) has gained attention, aiming to continuously acquire new semantic categories while preserving recognition capabilities for old ones. While exemplar replay (Rebuffi et al., 2017; Hou et al., 2019; Douillard et al., 2020; Wang et al., 2022) mitigates forgetting by replaying real old data, it raises significant concerns regarding privacy and storage. Exemplar-free CIL (EFCIL) avoids these issues through distillation (Li & Hoiem, 2017; Shi & Ye, 2023) or architectural (Ramesh & Chaudhari, 2022; Rypesc et al., 2024) constraints, yet typically underperforms replay-based methods. Generative replay presents a promising alternative by synthesizing old data. However, traditional generative replay methods require the continual training of generators (*e.g.*, GANs (Shin et al., 2017) or diffusion models (Gao & Liu, 2023)), leading to high computational costs and generator forgetting. While recent approaches attempt to update the generator via LoRA (Meng et al., 2024), they still require training multiple adapter modules per stage or class, resulting in mediocre training efficiency and increased parameter overhead.

[1] Huazhong University of Science and Technology, Wuhan, China [2] State Key Laboratory of Multispectral Information Intelligent Processing Technology, Wuhan, China [3] University of Science and Technology of China, Anhui, China [4] Wangxuan Institute of Computer Technology, Peking University, Beijing, China. Correspondence to: Xu Zou <zoux@hust.edu.cn>.

*Proceedings of the 43$^{rd}$ International Conference on Machine Learning*, Seoul, South Korea. PMLR 306, 2026. Copyright 2026 by the author(s).

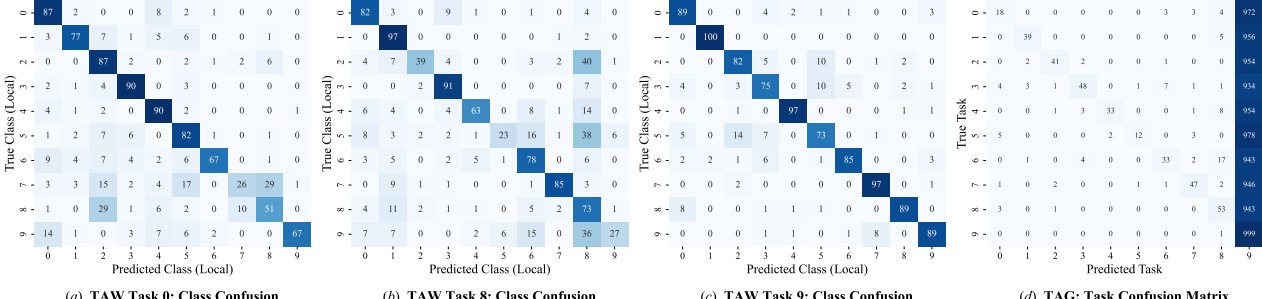

*Figure 2.* Visualization of the "Domain Shortcut" via Confusion Matrices. **TAW (a, b, c)**: Within-task classification remains accurate as class differences outweigh domain differences. **TAG (d)**: When mixing tasks, the model overwhelmingly misclassifies old-class real data (Tasks 0-8) as the current real task (Task 9), confirming that domain artifacts dominate semantic cues in the decision process.

Recent advances using synthetic data for model training have been used to enhance the performance of deep learning models (Tian et al., 2023; Fan et al., 2024; Jiang et al., 2025). Motivated by this, we explore using a training-free generator to synthesize old-class data for EFCIL. To generate high-quality data, we employ an Attribute-Aware Generative Replay Strategy: we extract detailed image captions using a Vision-Language Model and feed them to the training-free generator to synthesize old-class images that closely match the original object and background. **This approach appears to effectively address both the challenge of unavailable old data and the need for frequent training of traditional generators in EFCIL. But that is not the case.**

When applying this paradigm to incremental training—by mixing synthetic old-class data with real new-class data—we observe a significant degradation in Task-Agnostic Accuracy (TAG), while Task-Aware Accuracy (TAW) remains strong. **We attribute this issue not to insufficient semantic richness in synthetic data, but to domain bias in the feature space**. When real new data and synthetic old data are trained simultaneously, the synthetic data retains the semantic information necessary to resist forgetting. However, domain bias between real and synthetic data causes the model to also learn domain-specific features, leading it to mistakenly treat all real data as new tasks, resulting in a "**Domain Shortcut**". As shown in Figure 1(a), the "Domain Shortcut" arises because inter-domain discrepancies outweigh class discrepancies. While class boundaries are well-preserved within the same domain (maintaining high TAW), the large gap between real and synthetic domains collapses the global decision boundary. Consequently, real old-class samples are misclassified into new classes due to their domain attributes, which is further illustrated in Figure 2, where TAW remains high, but TAG collapses heavily. **These observations suggest that the primary bottleneck of generative replay is not data quality, but the "Domain Shortcut" caused by domain-induced feature bias.**

To address this problem, we propose DREAM (**D**omain-

**R**egularized **E**xemplar-free **A**lignment **M**odel), a feature rectification framework that explicitly eliminates the "Domain Shortcut". As shown in Figure 1(b), DREAM projects features into a semantic subspace via Feature Domain Rectification (FDR). By using Dynamic Domain Centering and Hierarchical Subspace Rectification, real and synthetic data are aligned in the feature subspace, compelling the model to classify based on semantic rather than domain cues. Additionally, Real-Anchored Prototype Consolidation (RAPC) aligns the semantic distributions of real and synthetic data, further reducing domain drift. These mechanisms enable stable and efficient EFCIL using a training-free generator.

We summarize our contributions as follows:

- We explore a training-free generative replay paradigm for EFCIL, enabling effective incremental learning without the computational overhead or parameter costs associated with training additional generators.

- We reveal the "**Domain Shortcut**" phenomenon in generative replay-based EFCIL as the main failure mode of training-free generators, causing models to exploit domain cues instead of semantic information.

- We propose DREAM, a feature rectification framework that explicitly excises "Domain Shortcut" caused by domain-dominant directions via Feature Domain Rectification (FDR) and aligns distributions via Real-Anchored Prototype Consolidation (RAPC).

- Extensive experiments on CIFAR-10, CIFAR-100, ImageNet-Subset, and Tiny-ImageNet demonstrate that DREAM outperforms existing methods, establishing a new insight for synthetic-data-assisted CIL.

## 2. Related Work

**Class-Incremental Learning.** Class-Incremental Learning (CIL) aims to continuously learn new classes without

forgetting old knowledge. A widely adopted method to mitigate catastrophic forgetting is exemplar replay (Rebuffi et al., 2017; Yan et al., 2021), which stores real old data, but raises privacy and storage concerns. As a result, Exemplar-Free CIL (EFCIL) has been actively researched (Zhu et al., 2021b; Petit et al., 2023), typically using knowledge distillation (Li & Hoiem, 2017; Shi & Ye, 2023) or model expansion (Rypesc et al., 2024) to preserve old knowledge. Alternatively, generative replay methods synthesize old data to approximate exemplar replay. Early works used GAN-based generators (Shin et al., 2017; Wu et al., 2018), while more recent methods employ diffusion models for improved stability and sample quality (Gao & Liu, 2023; Meng et al., 2024). However, most generative replay methods require retraining of generators, increasing training complexity and parameter costs. In contrast, our approach leverages the zero-shot generation capabilities of frozen generative models, synthesizing high-quality old data from rich text prompts, which eliminates the need for frequent generator retraining and addresses stringent data privacy concerns in EFCIL.

**Pre-trained Generative Models for Data Synthesis.** Pre-trained generative models exhibit strong data synthesis capabilities (Rombach et al., 2022) and have been widely adopted for data augmentation (Zhou et al., 2023b), long-tailed learning (Jiang et al., 2025), and few-shot learning (Ma et al., 2025). Previous work shows that synthetic data can enrich data distributions and improve downstream performance (Nguyen et al., 2023; Alimisis et al., 2025; Qiu et al., 2024). These methods typically assume a static learning setup where synthetic and real data coexist symmetrically and play similar semantic roles. However, in EFCIL, this paradigm introduces an asymmetry: old classes only have synthetic data, while real data correspond only to new classes. This disrupts learning dynamics, leading to a critical failure mode known as the "Domain Shortcut".

**Feature Alignment and Subspace Learning.** Domain Adaptation (DA) aligns distributions via statistical matching (Long et al., 2015) or adversarial learning (Ganin et al., 2016). However, standard DA typically requires concurrent access to source data, rendering it infeasible for EFCIL. Parallelly, subspace learning methods in CIL, such as Orthogonal Gradient Descent (Farajtabar et al., 2020) and Gradient Projection Memory (Saha et al., 2021), leverage orthogonal projections to minimize inter-class interference. Yet, these methods utilize orthogonality primarily to preserve knowledge rather than eliminate domain discrepancies. Unlike standard alignment approaches that assume static domain definitions, the generative replay setting presents a unique challenge where domain bias is structurally coupled with class increments. Addressing this, we diverge from gradient-based constraints and propose a geometric rectification strategy that explicitly identifies and excises domain-sensitive subspaces to enforce semantic invariance.

## 3. Theoretical Analysis

Due to the domain discrepancy between synthetic and real data, directly mixing generated old-class data with real new-class data for training induces a "Domain Shortcut", causing real old-class data to be misclassified as new tasks during inference. We formalize this phenomenon via feature decomposition. We assume that the feature $z \in \mathbb{R}^D$ extracted by $f_\theta$ decomposes into semantic and domain components: $z = z_{sem}(y) + z_{dom}(d) + \epsilon$, where $z_{sem}(y)$ encodes class semantics, $z_{dom}(d)$ captures domain bias with $d \in real, syn$, and $\epsilon$ denotes noise. In generative replay-based EFCIL, a structural **asymmetry** arises: old-class data are exclusively synthetic ($P(d = \text{syn} \mid y \in \mathcal{D}old) = 1$), while new-class data are purely real ($P(d = \text{real} \mid y \in \mathcal{D}new) = 1$).

**Proposition 3.1** (Domain Shortcut Phenomenon). *Define the domain bias vector as $\Delta_{dom} = \mathbb{E}[z_{dom}(real)] - \mathbb{E}[z_{dom}(syn)]$. When the domain variance significantly outweighs semantic variance ($|\Delta_{dom}|^2 \gg Var(z_{sem})$), the model prioritizes the $\Delta_{dom}$ direction, misclassifying all real data as a new task. Consequently, real test old-class data ($d = real$) are wrongly classified as the new tasks.*

**Proof** of Proposition 3.1 is seen in Appendix A.1.

Since all synthetic data are produced by the same frozen generator, they exhibit the following properties.

**Corollary 3.2** (Shared Domain Difference Subspace). *Since the generator is **training-free**, the statistical characteristics of the synthetic domain remain time-invariant across all classes. Similarly, the real images share a consistent natural domain distribution. Consequently, the domain gap $\mathcal{S}_{dom}$, arising from the discrepancy between these two stable distributions, is structurally consistent and shared. Formally:*

$$\forall c \in \mathcal{C}_{1:T}, \quad (\mathbb{E}[z|c, real] - \mathbb{E}[z|c, syn]) \in \mathcal{S}_{dom}. \quad (1)$$

**Analysis** of Corollary 3.2 is seen Appendix A.2.

**Corollary 3.3** (Structured Domain Shift vs. Isotropic Intra-class Variance). *Attributable to the fixed nature of the training-free generator, the synthesized data exhibits consistent stylistic tendencies. Consequently, the domain bias is not stochastic noise but comprises inherent systematic artifacts. This bias defines a **structured component** $\Sigma_{dom}$ that dominates specific directions in the feature space. Conversely, intra-class semantic variations (e.g., color, pose, background) are characterized as **statistically isotropic components** $\sigma_{sem}^2 I$. Therefore, the covariance of class-conditional residuals decomposes as:*

$$Cov(z|c) \approx \Sigma_{dom} + \sigma_{sem}^2 I, \quad s.t. \ \lambda_1(\Sigma_{dom}) \gg \sigma_{sem}^2. \quad (2)$$

**Analysis** of Corollary 3.3 is seen in Appendix A.3.

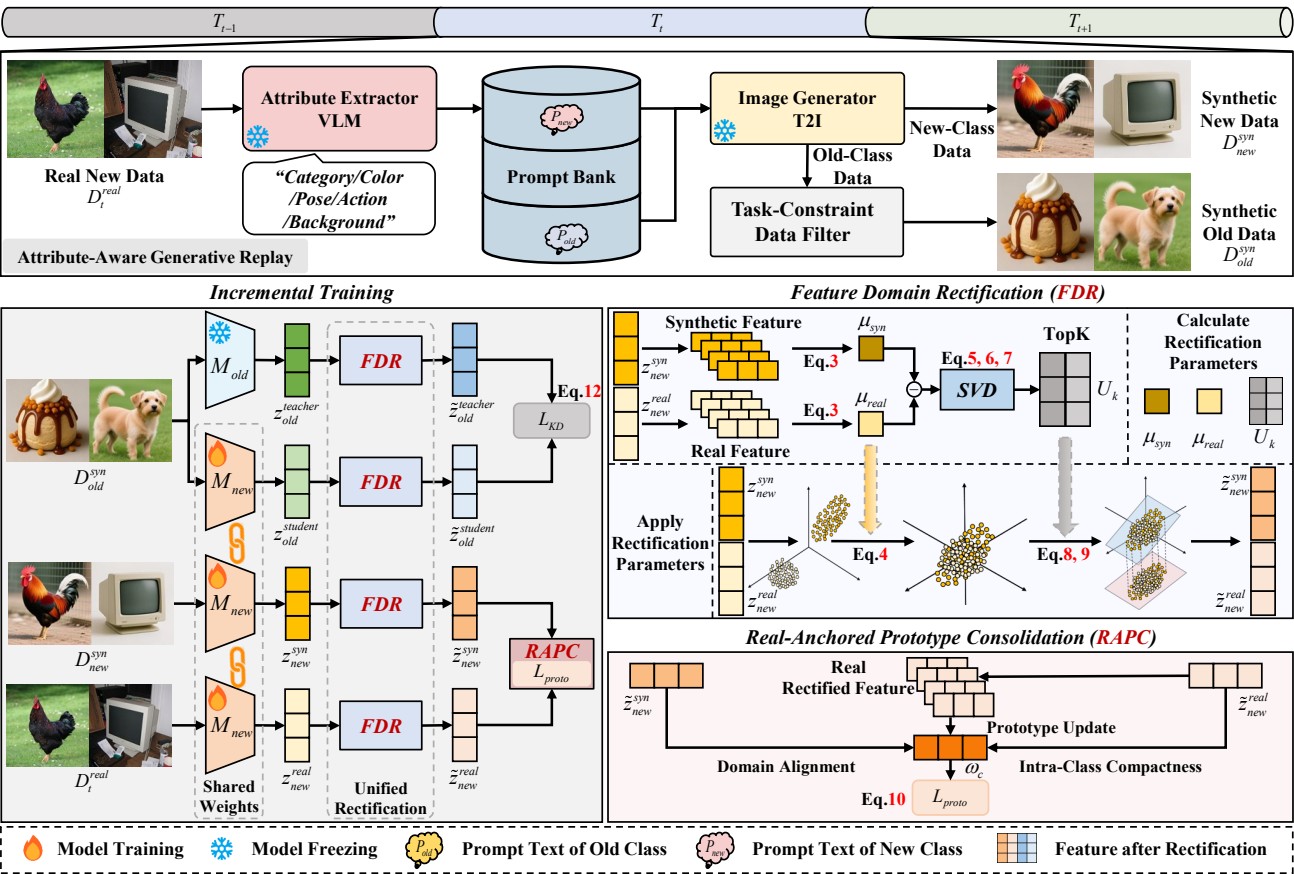

*Figure 3.* Overview of DREAM. The pipeline integrates Attribute-Aware Generative Replay (blue) with a novel training pipeline based on Feature Domain Rectification (yellow). **(1) Synthesis**: We utilize a VLM to extract rich attributes and a frozen generator to synthesize diverse old and real data. **(2) Rectification (FDR)**: We perform Dynamic Domain Centering and Hierarchical Subspace Rectification to explicitly remove domain-sensitive subspaces (visualized bottom-left). **(3) Alignment (RAPC)**: A Real-Anchored Prototype Consolidation loss aligns synthetic and real features with real-data anchors to ensure domain alignment and intra-class compactness.

These corollaries motivate estimating the domain subspace from paired new-class data, enabling the removal of real–synthetic domain gap across old and new classes.

## 4. Methodology

**Preliminaries.** We focus on EFCIL under strict privacy constraints. The learning process consists of $T$ sequential stages. At stage $t$, the model only has access to the real data of new classes for the current task, denoted as $\mathcal{D}_t^{real} = \{(x, y)|y \in \mathcal{C}_t\}$, without access to any real old-class data. To mitigate catastrophic forgetting, we introduce a frozen pre-trained text-to-image generator $G$ to synthesize data.

### 4.1. Attribute-Aware Generative Replay.

To obtain high-quality synthetic data, we use an Attribute-Aware Generative Replay strategy. We employ Qwen2.5-VL (Bai et al., 2025) to extract rich semantic descriptions

($T_y$) from real images, covering *category*, *color*, *pose*, *background*, and *action*, which are stored in a Prompt Bank.

**For Old Classes:** We retrieve prompts $T_{old}$ from the bank and feed them into $G$ to generate the synthetic old-class data $\mathcal{D}_{old}^{syn}$, applying "Task-Constraint Data Filtering" to ensure high data quality. **For New Classes:** We also use $G$ to generate synthetic data for new classes, $\mathcal{D}_{new}^{syn}$, to assist with subsequent feature rectification (Sec. 4.2 and 4.3). The details are seen in Algorithm 1. Consequently, the training set at stage $t$ comprises three parts: real new data $\mathcal{D}_{new}^{real}$, synthetic new data $\mathcal{D}_{new}^{syn}$, and synthetic old data $\mathcal{D}_{old}^{syn}$.

### 4.2. Feature Domain Rectification (FDR)

To block the shortcut, we must break the asymmetry by removing the influence of $z_{dom}(d)$ from the features. DREAM uses paired $\mathcal{D}_{new}^{real}$ and $\mathcal{D}_{new}^{syn}$ to explicitly estimate and eliminate the domain subspace, as shown in Figure 3. Since $\mathcal{D}_{new}^{real}$ and $\mathcal{D}_{new}^{syn}$ share the same semantic subspace but

belong to different domains, they provide an opportunity to isolate domain differences without semantic contamination.

Based on Corollary 3.2, we can use the projection matrix from the new classes and generalize it to the old classes.

**Dynamic Domain Centering.** We first eliminate the first-order statistic (mean) shift. We maintain the real mean $\mu_{real}$ and synthetic mean $\mu_{syn}$ of the current task's new classes using Exponential Moving Average before each epoch:

$$
\begin{aligned}
\mu_{real} &= \frac{1}{|\mathcal{C}_t|} \sum_{c \in \mathcal{C}_t} \frac{1}{N_c^{real}} \sum_{z \in \mathcal{D}_c^{real}} z, \\
\mu_{syn} &= \frac{1}{|\mathcal{C}_t|} \sum_{c \in \mathcal{C}_t} \frac{1}{N_c^{syn}} \sum_{z \in \mathcal{D}_c^{syn}} z.
\end{aligned}
\tag{3}
$$

We perform centering based on its source $d$ for any $z$, :

$$
z' = z - (\mathbb{I}_{[d=real]}\mu_{real} + \mathbb{I}_{[d=syn]}\mu_{syn}). \tag{4}
$$

It aligns the geometric centers of the two distributions to the origin, removing the translational component in $\Delta_{dom}$.

**Hierarchical Subspace Rectification.** To further excise higher-order discrepancies, we propose a hierarchical Singular Value Decomposition (SVD) strategy to identify and excise these domain-sensitive directions.

Corollary 3.3 ensures that the principal component extracted via SVD accurately locks onto the consistent domain drift direction, rather than capturing semantic variations.

*Class-wise Drift Mining (Micro-SVD).* To accurately extract domain difference directions, we need to eliminate the interference of $z_{sem}(y)$. For each category $c \in \mathcal{C}_t$ in the new task, we first compute the center of synthetic data $\bar{\mu}_c^{syn} = \mathbb{E}_{x \sim \mathcal{D}_{new}^{syn}|c}[f(x)]$. Then, we calculate the residual of the real sample $z_{i,c}^{real}$ relative to the synthetic center:

$$
\begin{aligned}
r_{i,c} &= z_{i,c}^{real} - \bar{\mu}_c^{syn} \\
&\approx \underbrace{(z_{dom}^{real} - z_{dom}^{syn})}_{\text{Consistent Domain Shift}} + \underbrace{(z_{sem,i} - \bar{z}_{sem})}_{\text{Intra-class Var}} + \underbrace{(\epsilon_i - \bar{\epsilon})}_{\text{Noise}}.
\end{aligned}
\tag{5}
$$

Then, we perform SVD on the set $R_c = \{r_{i,c}\}_{i=1}^{N_c}$:

$$
R_c = U_c \Sigma_c V_c^T. \tag{6}
$$

According to Corollary 3.3, the residual energy concentrates along the domain direction and is dispersed across semantic dimensions. Thus, we extract the first principal component $v_c$, which captures the dominant domain drift while suppressing isotropic semantic variations and random noise.

*Shared Subspace Aggregation (Macro-SVD).* To obtain a universal domain subspace applicable to all classes (including unseen old classes) under the current model, we aggregate the drift directions of all new classes into a matrix

$M = [v_{c_1}, v_{c_2}, \ldots, v_{c_{|\mathcal{C}_t|}}]$ and perform a second SVD:

$$
M = U_{all} \Sigma_{all} V_{all}^T. \tag{7}
$$

We select the top $k$ column vectors of $U_{all}$ to form the basis matrix $U_k \in \mathbb{R}^{D \times k}$. According to Corollary 3.2, this shared subspace $\mathcal{S}_{dom}$ can generalize to cover the domain bias of all new and old class data.

*Soft Orthogonal Projection.* To preserve semantic integrity while mitigating domain bias, we construct a soft projection:

$$
P = I - \gamma U_k U_k^T, \tag{8}
$$

In the high-dimensional feature space of deep networks, domain and semantics often exhibit non-trivial coupling rather than perfect orthogonality. Forcing a complete projection of features onto the orthogonal complement of $\mathcal{S}_{dom}$ might discard semantic information that is structurally embedded within the domain-sensitive subspace, leading to a drop in accuracy. Therefore, we adopt a soft projection to suppress the magnitude of domain artifacts rather than eliminating them so that it no longer dominates the distance metric, effectively enhancing the feature's signal-to-noise ratio (SNR) without disrupting the underlying semantic manifold. The final rectified feature is:

$$
\tilde{z} = Pz'. \tag{9}
$$

Through this projection, the feature space is mapped to a "Domain-Agnostic Manifold", compelling the model to rely on the preserved semantic features $z_{sem}$ for decision-making, thereby breaking the "Domain Shortcut".

### 4.3. Real-Anchored Prototype Consolidation (RAPC)

Although FDR aligns the feature space, the semantic distribution of synthetic data may still drift, and real data might become loose due to rectification. To further align domains and enhance intra-class compactness, we propose a Real-Anchored Prototype Consolidation strategy. We maintain a set of "Real Prototypes" $\Omega = \{\omega_c\}_{c=1}^{|\mathcal{C}|}$, updated using Exponential Moving Average of the rectified features of real new-class data. According to Corollary 3.2, the old-class domain is also optimized. The loss function is defined as:

$$
\begin{aligned}
\mathcal{L}_{Proto} &= 1 - \underbrace{\frac{1}{|B_{syn}|} \sum_{(x,y) \in B_{syn}} cos(\tilde{z}, \omega_y)}_{\text{Term A: Domain Alignment}} \\
&+ 1 - \underbrace{\frac{1}{|B_{real}|} \sum_{i \in B_{real}} cos(\tilde{z}, \omega_y)}_{\text{Term B: Intra-Class Compactness}}.
\end{aligned}
\tag{10}
$$

**Term A** forces synthetic data to move closer to the real prototype, patching residual domain gaps. **Term B** ensures that the real data itself remains compact.

*Table 1.* **Comparison of Task-agnostic Average Incremental Accuracy (%) across four benchmarks.** We compare DREAM with exemplar-free (*e.g.*, SEED), exemplar-based (*e.g.*, T-CIL), model inversion (*e.g.*, R-DFCIL), and generative replay methods (*e.g.*, DiffClass). DREAM consistently achieves the best performance across all datasets and settings among exemplar-free approaches, while attaining performance comparable to exemplar-based methods. Best results are shown in **bold**, and second-best results are underlined. Missing results are denoted by "–", as some methods do not report the corresponding settings or do not release their implementations.

| Type | Methods | CIFAR-10 | | CIFAR-100 | | | ImageNet-Subset | | | TinyImageNet | | |
|---|---|---|---|---|---|---|---|---|---|---|---|---|
| | | T=2 | T=5 | T=5 | T=10 | T=20 | T=5 | T=10 | T=20 | T=5 | T=10 | T=20 |
| **Baselines** | Joint | 93.27 | 91.05 | 73.61 | 70.45 | 71.45 | 80.46 | 77.13 | 76.41 | 60.84 | 58.55 | 54.34 |
| | Fine-Tuning | 70.28 | 42.84 | 35.12 | 24.72 | 15.89 | 38.44 | 26.29 | 17.18 | 28.21 | 20.06 | 12.82 |
| **Exemplar-Free** | EWC (Kirkpatrick et al., 2017) | 72.10 | 42.24 | 39.95 | 25.29 | 17.05 | 41.36 | 27.74 | 17.38 | 29.61 | 20.47 | 14.10 |
| | LwF (Li & Hoiem, 2017) | 79.94 | 56.28 | 41.71 | 30.69 | 19.87 | 52.55 | 39.27 | 26.87 | 32.02 | 22.47 | 15.26 |
| | PASS (Zhu et al., 2021b) | 63.72 | 59.75 | 63.31 | 52.01 | 41.84 | 55.75 | 33.75 | 27.30 | 39.55 | 30.03 | 18.64 |
| | IL2A (Zhu et al., 2021a) | 63.42 | 51.66 | 58.67 | 43.28 | 40.54 | 62.66 | 43.46 | 35.59 | 36.56 | 32.38 | 16.70 |
| | SSRE (Zhu et al., 2022) | 62.84 | 51.52 | 56.96 | 43.41 | 31.07 | 52.25 | 46.00 | 34.96 | 33.23 | 28.82 | 16.22 |
| | FeTrIL (Petit et al., 2023) | 83.47 | 66.59 | 58.68 | 47.14 | 37.25 | 58.40 | 46.44 | 37.64 | 40.46 | 30.77 | 23.68 |
| | SEED (Rypesc et al., 2024) | 87.89 | 74.03 | 63.05 | 62.04 | 57.42 | 69.08 | 67.55 | 62.26 | 51.16 | 48.77 | 39.68 |
| **Exemplar-Based** | T-CIL (Hwang et al., 2025) | - | 65.86 | - | 56.25 | - | - | - | - | - | 31.48 | - |
| | T-CIL+DER (Hwang et al., 2025) | - | 74.93 | - | 69.98 | - | - | - | - | - | 47.79 | - |
| **Exemplar-Free Generative Replay** | ABD (Smith et al., 2021) | 84.07 | 72.12 | 60.78 | 54.00 | 43.32 | 67.12 | 57.06 | 45.75 | 45.80 | 41.59 | 35.65 |
| | R-DFCIL (Gao et al., 2022) | 85.90 | 74.98 | 64.67 | 59.18 | 49.76 | 68.42 | 59.36 | 49.99 | 49.36 | 44.54 | 39.52 |
| | DiffClass (Meng et al., 2024) | - | - | 69.77 | 68.05 | 67.10 | 74.85 | 73.87 | 72.51 | - | - | - |
| | AHR (Nori et al., 2025) | - | 77.12 | - | 54.43 | - | - | - | - | - | - | - |
| | **DREAM (Ours)** | **91.08** | **88.98** | **71.55** | **69.73** | **68.48** | **78.37** | **76.57** | **74.43** | **55.54** | **52.92** | **51.00** |

### 4.4. Optimization Objective

The model is optimized in the *rectified* feature space. The total loss $\mathcal{L}$ is defined as:

$$\mathcal{L} = \mathcal{L}_{CE} + \lambda_{KD}\mathcal{L}_{KD} + \lambda_{Proto}\mathcal{L}_{Proto}. \quad (11)$$

$\mathcal{L}_{CE}$ is the Cross-Entropy loss computed on $\mathcal{D}_t$, $\mathcal{D}_{syn\_new}$ and $\mathcal{D}_{syn\_old}$. Crucially, we employ **Rectified Feature Distillation** to prevent forgetting:

$$\mathcal{L}_{KD} = 1 - cos(\tilde{z}^{student}, \tilde{z}^{teacher}). \quad (12)$$

Both student and teacher features are rectified using their respective stage-specific projectors, ensuring knowledge transfer robust to domain shifts across incremental stages.

## 5. Experiments

### 5.1. Experimental Setup

**Datasets.** To evaluate the effectiveness of DREAM and ensure fair comparison, we conduct experiments on four widely adopted benchmark datasets in CIL: CIFAR-10, CIFAR-100 (Krizhevsky et al., 2009), ImageNet-Subset (Deng et al., 2009), and TinyImageNet (Le & Yang, 2015). CIFAR-10 and CIFAR-100 consist of $32 \times 32$ images with 10 and 100 classes, respectively. Both datasets contain 50,000 training images and 10,000 test images. ImageNet-Subset-100 is constructed by selecting 100 classes from ImageNet-1k based on a fixed random seed (1993). TinyImageNet comprises 200 classes with $64 \times 64$ images, which includes 100,000 training samples and 10,000 test samples.

**Incremental Setting.** Unlike the setting in (Zhang et al., 2024; Liu et al., 2020), where the first task contains more classes than subsequent tasks, we evaluate in a broadly adopted and more challenging equal-split setting, where each task has the same class number, as in established EF-CIL protocols (Gao et al., 2022; Rypesc et al., 2024; Meng et al., 2024). For CIFAR-10, we split into $T = 2$ or 5 tasks. For CIFAR-100, ImageNet-Subset, and TinyImageNet, we split the classes equally into $T = 5$, 10, or 20 tasks. For fairness, all methods are evaluated using the same fixed class order. Following previous works (Petit et al., 2023; Rypesc et al., 2024), we report the Task-agnostic **Average Incremental Accuracy** ($\mathcal{A}_{avg}$):

$$\mathcal{A}_{avg} = \frac{1}{T}\sum_{t=1}^{T} \mathcal{A}_t, \quad (13)$$

defined as the mean of the accuracy achieved after each incremental phase, and $\mathcal{A}_t$ denotes the classification accuracy on the test set of all seen classes after learning task $t$.

**Implementation Details.** Following previous works (Gao et al., 2022; Rypesc et al., 2024; Meng et al., 2024), we employ a modified **32-layer ResNet** (He et al., 2016) as the backbone for CIFAR-10/100 and TinyImageNet and use the standard **ResNet-18** for ImageNet-Subset. The comparative results are from exiting paper or reproducing results using FACIL (Masana et al., 2022) and PyCIL (Zhou et al., 2023a) benchmarks. All models are trained from scratch. Regarding the generative replay framework, we utilize the frozen **Qwen-Image** (Wu et al., 2025) as image generators, with text prompts derived from **Qwen2.5-VL** (Bai et al., 2025).

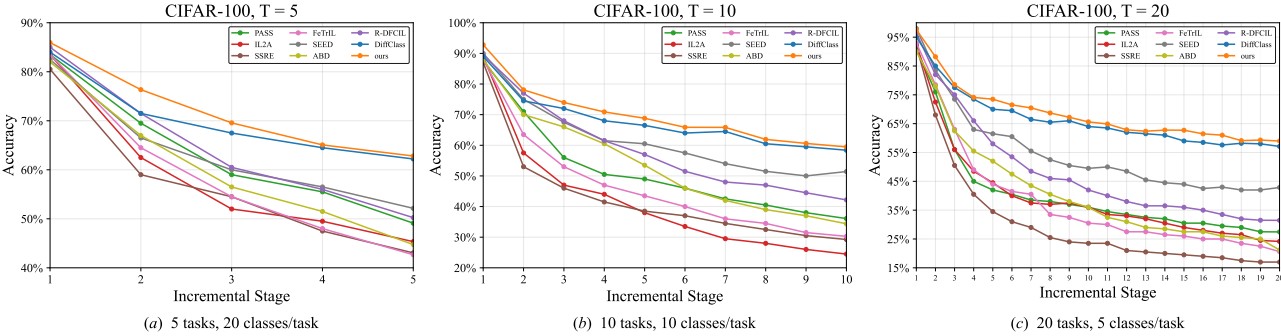

*Figure 4.* Task-agnostic average accuracy curves on CIFAR-100. Results for (a) $T = 5$ tasks, (b) $T = 10$ tasks, and (c) $T = 20$ tasks show that DREAM (orange curve) consistently outperforms state-of-the-art exemplar-free methods.

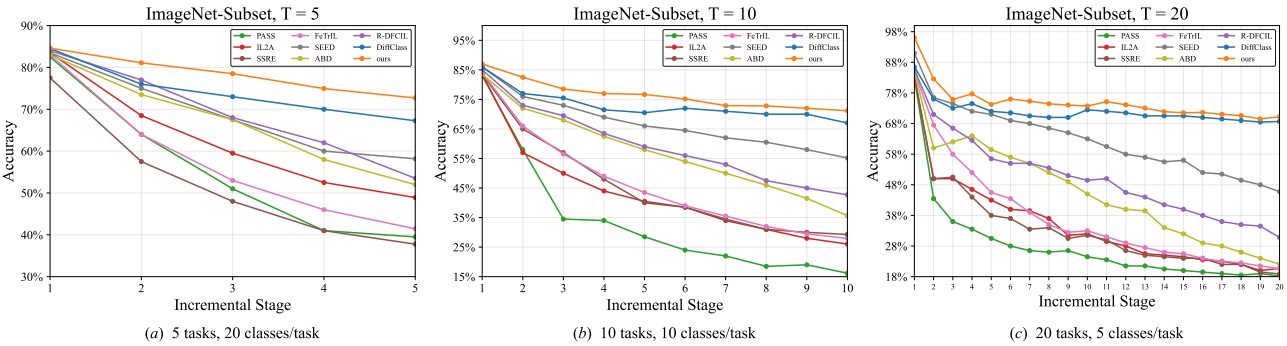

*Figure 5.* Task-agnostic average accuracy curves on ImageNet-Subset. Results for (a) $T = 5$ tasks, (b) $T = 10$ tasks, and (c) $T = 20$ tasks demonstrate that DREAM (orange curve) consistently outperforms state-of-the-art exemplar-free methods.

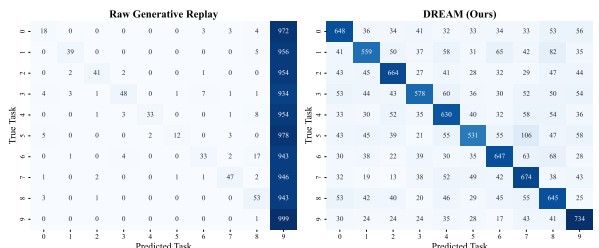

*Figure 6.* Confusion Matrices on CIFAR-100. **Left**: The baseline exhibits a severe "vertical" bias, misclassifying old data as the new task due to the "domain shortcut". **Right**: DREAM eliminates this bias, restoring the diagonal structure and ensuring reliance on semantic cues, effectively breaking the "Domain Shortcut".

## 5.2. Results and Analysis

We evaluate DREAM across 4 benchmark datasets under Exemplar-Free, Exemplar-Based, and Generative Replay CIL settings. As shown in Table 1, DREAM outperforms all baselines in the EFCIL setting and achieves results comparable to the exemplar-based method.

**Analysis on CIFAR-10.** DREAM achieves near-perfect performance on CIFAR-10 in both $T = 2$ and $T = 5$ settings. Notably, at $T = 5$, DREAM surpasses SEED by 14.95% (74.03% for SEED), highlighting the critical role of the "Domain Shortcut" in simpler tasks. Our rectification method effectively addresses this challenge, confirming its efficacy on simpler distributions.

**Analysis on CIFAR-100.** On CIFAR-100, DREAM outperforms all methods across varying task sequences. In the 5/10/20-task settings, our method achieves 71.55%, 69.73%, 68.48%, surpassing DiffClass (69.77%, 68.05%, 67.10%) and R-DFCIL (64.67%, 59.18%, 49.76%). Furthermore, DREAM exhibits a massive advantage over traditional EFCIL methods like SEED (63.05%, 62.04%, 57.42%). As illustrated in Figure 4, DREAM maintains a robust trajectory throughout the process, demonstrating stability and effective error mitigation, especially in long-sequence scenarios.

**Analysis on ImageNet-Subset.** On ImageNet-Subset ($224 \times 224$), DREAM shows superior robustness to both traditional EFCIL and generative methods. In the 5/10/20-task settings, our method achieves 78.37%, 76.57%, 74.43%, outperforming DiffClass (74.85%, 73.87%, 72.51%) and R-DFCIL (68.42%, 59.36%, 49.99%). The performance

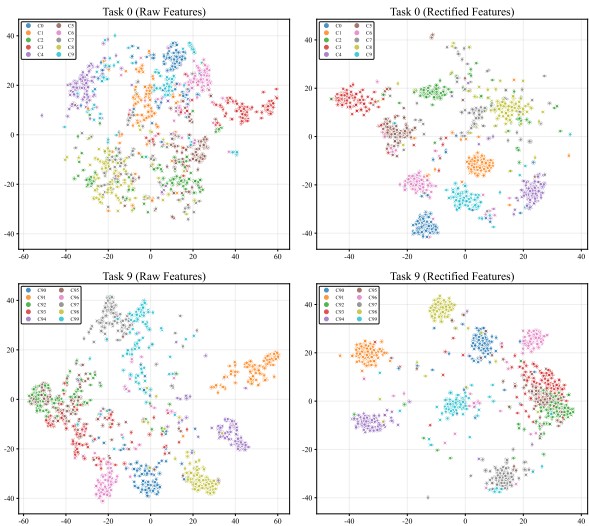

*Figure 7.* Intra-task t-SNE Visualization on CIFAR-100. We visualize feature spaces for Task 0 (top) and Task 9 (bottom). The left column (Raw) shows dispersed distributions, while the right column (Rectified) shows more compact and clustered representations, validating that our rectified distillation enhances intra-class compactness and semantic separability.

*Table 2.* Component Effectiveness Analysis. We incrementally add Centering, HSR, and RAPC to the baseline (the same data and just use $\mathcal{L}_{CE}$ and $\mathcal{L}_{KD}$) to validate the contribution of each module.

| Centering | HSR | RAPC | Acc (%) |
|:---:|:---:|:---:|:---:|
| Baseline (Raw Generative Replay) | | | 33.27 |
| ✓ | | | 41.02 |
| | ✓ | | 58.46 |
| ✓ | ✓ | | 67.69 |
| ✓ | ✓ | ✓ | **69.73** |

gap widens significantly against traditional EFCIL methods like SEED (62.26%), surpassing by 12.17% at $T = 20$. As shown in Figure 5, while traditional EFCIL methods exhibit a steep decline due to the inability to rehearse old patterns, DREAM follows a much flatter trajectory. This confirms the synergy of our framework: the data synthesized by the training-free generator ensures high-fidelity semantic coverage, while DREAM effectively rectifies the domain bias in complex high-resolution images.

**Analysis on Tiny-ImageNet.** On Tiny-ImageNet with 200 classes, DREAM achieves 55.54%, 52.92%, 51.00% accuracy, outperforming R-DFCIL (49.36%, 44.54%, 39.52%) and SEED (51.16%, 48.77%, 39.68%). This validates that DREAM is also effective in dense semantic spaces, preserving distinct decision boundaries where traditional EFCIL methods fail to retain class discriminability.

### 5.3. Visualization Analysis

To intuitively validate our framework, we analyze classification behavior and feature quality on CIFAR-100 ($T = 10$).

*Table 3.* Sensitivity Analysis of FDR Hyperparameters. We evaluate the impact of rectification strength $\gamma$ and subspace dimension $k$ on model performance. Default settings are bolded.

*(a)* Rectification Strength $\gamma$

| $\gamma$ | Acc (%) |
|:---:|:---:|
| 0.6 | 65.92 |
| 0.7 | 68.06 |
| **0.8** | **69.73** |
| 0.9 | 65.59 |
| 1.0 | 56.92 |

*(b)* Subspace Dimension $k$

| $k$ | Acc (%) |
|:---:|:---:|
| 1 | 66.74 |
| 3 | 67.96 |
| **5** | **69.73** |
| 8 | 64.68 |
| 10 | 62.02 |

*Table 4.* Sensitivity Analysis of Loss Weights. We assess the stability of the model under varying weights for prototype loss ($\lambda_{Proto}$) and distillation loss ($\lambda_{KD}$).

*(a)* Proto Weight $\lambda_{Proto}$

| $\lambda_{Proto}$ | Acc (%) |
|:---:|:---:|
| 0 | 67.69 |
| 0.05 | 68.76 |
| **0.10** | **69.73** |
| 0.50 | 68.52 |
| 1.00 | 68.02 |

*(b)* KD Weight $\lambda_{KD}$

| $\lambda_{KD}$ | Acc (%) |
|:---:|:---:|
| 0.1 | 66.61 |
| 0.5 | 68.60 |
| **1.0** | **69.73** |
| 5.0 | 68.25 |
| 10.0 | 63.78 |

As shown in the confusion matrices (Figure 6), the baseline (left) exhibits a severe "vertical" bias, misclassifying old real samples as part of the current real task, confirming the presence of the "Domain Shortcut." In contrast, DREAM (right) restores the diagonal structure, eliminating the domain bias and addressing the "Domain Shortcut". Additionally, Figure 7 presents the t-SNE visualization for tasks 0 and 9. Raw features (left) show dispersed distributions with loose boundaries for both old and new classes within a task, while DREAM (right) produces much more compact and clustered representations. This enhanced compactness demonstrates that our method effectively enforces stricter semantic separability within tasks.

### 5.4. Ablation Study

**Component Effectiveness.** In Table 2, we validate the contribution of each module by adding them to the raw generative replay baseline (just use $\mathcal{L}_{CE}$ and $\mathcal{L}_{KD}$) on CIFAR-100 ($T = 10$). The baseline yields poor accuracy (33.27%), confirming that the domain shortcut causes severe optimization conflicts. Aligning first-order statistics via "Centering" improves accuracy by 7.75%. Introducing HSR to reduce high-variance domain directions further increases accuracy to 67.69%, supporting our hypothesis that the domain shortcut exists in specific subspaces. Finally, RAPC refines the domain alignment and ensures compactness, reaching 69.73%.

**Hyperparameter Sensitivity.** We investigate the impact of Feature Domain Rectification (FDR) settings in Table 3. Regarding rectification strength, a "Soft Projection" ($\gamma = 0.8$)

*Table 5.* Impact of generator and filtering. DREAM remains effective across different generators, and filtering further improves performance by removing semantically inconsistent generations.

| Generator | Filtering | Data Vol. | Acc. (%) |
|---|---|---|---|
| SD1.5 (Rombach et al., 2022) | w/o Filter | 500 | 68.08 |
|  | w/ Filter | 425 | 68.84 |
| Qwen-Image (Wu et al., 2025) | w/o Filter | 500 | 68.67 |
|  | w/ Filter | 468 | **69.73** |

*Table 6.* Robustness to multi-source generators. DREAM consistently improves over raw generative replay even when replay data are synthesized from heterogeneous generators.

| Type | Generator Source | w/o DREAM | DREAM |
|---|---|---|---|
| Multi-source | 50% SD1.5 + 50% Qwen | 37.92 | 68.53 |
|  | 50% Qwen + 50% SD1.5 | **40.37** | 67.87 |
|  | Random mix | 39.33 | 67.77 |
|  | Avg | 39.21 | 68.06 |
| Single-source | SD1.5 | 32.65 | 68.84 |
|  | Qwen-Image | 33.27 | **69.73** |
|  | Avg | 32.96 | 69.29 |

achieves the optimal trade-off; notably, "Hard Projection" ($\gamma = 1.0$) causes a sharp performance drop to 56.92%, suggesting that the domain-sensitive subspace contains entangled semantic information that should be suppressed rather than deleted. Additionally, the method proves robust to the choice of subspace dimension $k$, peaking at $k = 5$.

We also analyze the sensitivity of the loss components $\lambda_{Proto}$ and $\lambda_{KD}$ in Table 4. Model performance remains stable across a reasonable range of weights, with $\lambda_{Proto} = 0.1$ and $\lambda_{KD} = 1.0$ yielding the highest accuracy (69.73%). This combination effectively aligns domains, ensuring sufficient knowledge transfer from the teacher model.

**Robustness to Generator and Filtering.** We further examine whether the performance of DREAM depends on the choice of generator or the filtering strategy. As shown in Table 5, DREAM achieves comparable performance with SD1.5 and Qwen-Image, obtaining 68.84% and 69.73% after filtering, respectively. This small gap indicates that the improvement mainly comes from DREAM rather than the use of a stronger generator. Filtering consistently improves both generators by removing low-quality or semantically inconsistent data. Notably, DREAM with SD1.5 still achieves 68.84%, which is competitive with the Qwen-Image setting and validates that our method is robust to the generator and does not rely on a specific high-fidelity generator. Some extra analyses are presented in the Appendix D.7.

**Robustness to Multi-Source Generators.** To examine whether DREAM depends on a single-source generator, we conduct heterogeneous replay experiments on CIFAR-100 ($T = 10$) by mixing SD1.5 and Qwen-Image with a 50/50 class split. We consider three mixed-source settings: the first half of classes generated by SD1.5 and the second half by Qwen-Image, the reverse split, and a random 50/50 class split.. As shown in Table 6, DREAM consistently improves multi-source replay from 39.21% to 68.06%. Interestingly, raw multi-source replay performs better than raw single-source replay (39.21% vs. 32.96%), suggesting that generator diversity can partially dilute fixed synthetic artifacts, but the performance remains poor without explicit rectification. After applying DREAM, single-source replay is slightly better than multi-source replay (69.29% vs. 68.06%), indicating that coherent artifacts from one frozen generator form a more compact domain subspace that is easier to remove via SVD, while heterogeneous generators introduce multi-modal biases. Nevertheless, the small 1.23% drop under multi-source replay demonstrates the robustness of DREAM to heterogeneous generator sources.

## 6. Conclusion

In this work, we identify the "Domain Shortcut" as a primary bottleneck in training-free generative replay EFCIL. To address this, we propose **DREAM** (**D**omain-**R**egularized **E**xemplar-free **A**lignment **M**odel) that integrates **Feature Domain Rectification (FDR)** and **Real-Anchored Prototype Consolidation (RAPC)**. By explicitly excising domain-sensitive subspaces and aligning synthesis and real features, DREAM effectively break the "Domain Shortcut" and mitigates catastrophic forgetting. Extensive experiments on CIFAR-10, CIFAR-100, ImageNet-Subset, and Tiny-ImageNet demonstrate that our method outperforms SOTA baselines. Future work will explore adapting the generator's embeddings to further enhance diversity for classes.

## Acknowledgements

This work was supported by the National Natural Science Foundation of China under Grant U24B20139 and by the Hubei Provincial Natural Science Foundation of China under Grant 2026AFA040.

The computation was completed on the HPC Platform at Huazhong University of Science and Technology.

## Impact Statement

This paper presents work whose goal is to advance privacy-preserving Class-Incremental Learning. There are many potential societal consequences of our work, including enabling secure AI evolution in privacy-sensitive domains (e.g., healthcare) and promoting Green AI by eliminating generator retraining costs; however, we emphasize the need for vigilance regarding the potential propagation of societal biases inherent in pre-trained foundation models.

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

# A. Theoretical Proofs and Empirical Verification

In this section, we provide rigorous mathematical proofs for the propositions and corollaries presented in the main manuscript. Furthermore, we provide empirical evidence to validate these theoretical corollaries, ensuring that our geometric interpretation of the "Domain Shortcut" holds in practice.

## A.1. Proof of Proposition 3.1: The Domain Shortcut Mechanism

**Proposition 3.1 Restated.** *Define the domain bias vector as $\Delta_{dom} = \mathbb{E}[z_{dom}(real)] - \mathbb{E}[z_{dom}(syn)]$. When the domain variance significantly outweighs semantic variance ($|\Delta_{dom}|^2 \gg Var(z_{sem})$), the model prioritizes the $\Delta_{dom}$ direction, misclassifying all real data as a new task. Consequently, real test old-class data ($d = real$) are wrongly classified as the new tasks.*

**Proof.** We verify this proposition by analyzing the gradient descent dynamics and inference behavior under two common classifier architectures: the Linear Classifier and the Cosine Classifier.

### A.1.1. CASE I: LINEAR CLASSIFIER

**1. Problem Setup.** Consider a simplified binary classification task at incremental stage $t$ with classes $y \in \{-1, +1\}$, where $y = -1$ represents Old Classes (Synthetic) and $y = +1$ represents New Classes (Real). According to the **Feature Decomposition Hypothesis** (Eq. 1 in the main manuscript), the feature $z$ is a superposition of semantic and domain components:

$$z = z_{sem}(y) + z_{dom}(d). \tag{14}$$

For the binary case, we define the unit semantic direction as $v_s$ and the unit domain direction as $v_d$. The domain bias vector corresponds to $\Delta_{dom} \approx 2 \cdot v_d$ (difference between real and synthetic), and the semantic margin corresponds to $\Delta_{sem} \approx 2 \cdot v_s$. Due to the generator-based replay setting (Asymmetry), the training data exhibits perfect correlation between label $y$ and domain $d$:

$$z_{train} = y \cdot v_s + y \cdot v_d = y(v_s + v_d). \tag{15}$$

A linear classifier computes the logit $s = w^T z$. The logistic loss is $\mathcal{L}(w) = \log(1 + \exp(-yw^T z))$.

**2. Gradient Descent Update.** We compute the gradient of the loss $\mathcal{L}$ with respect to the weight vector $w$:

$$\nabla_w \mathcal{L} = \frac{\partial \mathcal{L}}{\partial s} \frac{\partial s}{\partial w} = \frac{-y \exp(-yw^T z)}{1 + \exp(-yw^T z)} z = -(1 - p)yz, \tag{16}$$

where $p$ is the predicted probability of the true class. The update rule with learning rate $\eta$ at step $k$ is:

$$w^{(k+1)} \leftarrow w^{(k)} - \eta \nabla_w \mathcal{L} = w^{(k)} + \eta(1 - p)yz. \tag{17}$$

Substituting $z_{train} = y(v_s + v_d)$, the accumulated update direction becomes:

$$\Delta w \propto y \cdot [y(v_s + v_d)] = y^2(v_s + v_d) = v_s + v_d. \tag{18}$$

After convergence, the weight vector $w$ aligns with the summation of feature components:

$$w^* \approx c \cdot (v_s + v_d), \quad \text{for some } c > 0. \tag{19}$$

**3. Inference Failure.** Consider a **Real Test Sample** from an **Old Class**:

- True Label: $y_{test} = -1$ (Old).

- Domain: $d_{test} = +1$ (Real).

- Feature: $z_{test} = (-1)v_s + (+1)v_d = v_d - v_s$.

The prediction score is:

$$s_{test} = (w^*)^T z_{test} \propto (v_s + v_d)^T (v_d - v_s). \tag{20}$$

Expanding the dot product (assuming orthogonality $v_s^T v_d = 0$ as per Corollary 3.3):

$$s_{test} \propto \|v_d\|^2 - \|v_s\|^2 = \frac{1}{4}(\|\Delta_{dom}\|^2 - \|\Delta_{sem}\|^2). \tag{21}$$

**Conclusion:** If the domain variance significantly outweighs the semantic variance (*i.e.*, $\|\Delta_{dom}\| \gg \|\Delta_{sem}\|$), then $s_{test} > 0$. The model predicts $y = +1$ (New Class), ignoring the semantic cue.

### A.1.2. CASE II: COSINE CLASSIFIER

**1. Problem Setup.** Modern CIL methods use Cosine Classifiers to mitigate magnitude bias. The logit is $s = \tau \frac{w^T z}{\|w\|\|z\|}$. For analysis, we assume normalized inputs ($\|z\| = 1$) and weights ($\|w\| = 1$). The loss is $\mathcal{L} = -\log \frac{e^{\tau w_y^T z}}{\sum_j e^{\tau w_j^T z}}$.

**2. Gradient Descent Update.** The gradient for the target class weight $w_y$ is:

$$\nabla_{w_y} \mathcal{L} = -(1 - p_y)\tau(z - (w_y^T z)w_y). \tag{22}$$

The update pushes $w_y$ towards $z$ while maintaining normalization. Over training epochs, the weight vector for the New Class ($y = +1$) converges to the mean direction of its training samples:

$$w_{new} \propto \frac{v_s + v_d}{\|v_s + v_d\|}. \tag{23}$$

Similarly, for the Old Class ($y = -1$), $w_{old} \propto \frac{-v_s - v_d}{\|-v_s - v_d\|}$.

**3. Inference Failure.** Consider the same Real Old Sample: $z_{test} = v_d - v_s$. We compare the cosine similarity scores for New ($S_{new}$) and Old ($S_{old}$) classes.

$$S_{new} = w_{new}^T z_{test} \propto (v_s + v_d)^T(v_d - v_s) = \|v_d\|^2 - \|v_s\|^2.$$
$$S_{old} = w_{old}^T z_{test} \propto (-v_s - v_d)^T(v_d - v_s) = -(\|v_d\|^2 - \|v_s\|^2). \tag{24}$$

**Conclusion:** Even with normalization, the sign of the score is determined by $\|v_d\|^2 - \|v_s\|^2$ (which is proportional to $\|\Delta_{dom}\|^2 - \|\Delta_{sem}\|^2$). If $\|\Delta_{dom}\| \gg \|\Delta_{sem}\|$, then $S_{new} > 0$ and $S_{old} < 0$. The Cosine Classifier also assigns higher similarity to the New Class weight because the test sample shares the strong "Real Domain" component ($\Delta_{dom}$) with it.

### A.1.3. CONCLUSION AND EMPIRICAL VERIFICATION

**Theoretical Conclusion:** Both derivations confirm that when the domain gap dominates ($\|v_{dom}\| \gg \|v_{sem}\|$), the classifier—whether Linear or Cosine—establishes a decision boundary based on domain artifacts rather than semantic content. The Linear classifier fails due to magnitude bias, while the Cosine classifier fails due to angular domain alignment.

**Empirical Verification:** We provide empirical evidence to support this proof in Figure 8. The heatmap visualizes the Task-Agnostic Accuracy (TAG) classification flow for Task 0/1/8/9 data (Real) after training Task 9. Consistent with our derivation, the heatmap shows a massive concentration of predictions in the Task 9 region (T9/C#). The model correctly identifies the domain (Real) but incorrectly associates it with the only Real classes it currently knows (Task 9), verifying the existence of the Domain Shortcut. □

## A.2. Analysis and Verification of Corollary 3.2: Shared Domain Subspace

**Corollary 3.2 Restated.** *We assume that the domain bias introduced by the training-free generator is not randomly distributed in the feature space but is concentrated in a subspace $\mathcal{S}_{dom}$, and this subspace is shared across different classes (either new or old). That is:*

$$\forall c \in \mathcal{C}_{1:T}, \quad (\mathbb{E}[z|c, real] - \mathbb{E}[z|c, syn]) \in \mathcal{S}_{dom}. \tag{25}$$

### A.2.1. THEORETICAL ANALYSIS: THE SYSTEMATIC ARTIFACT HYPOTHESIS

While strict mathematical proof depends on the black-box nature of the generator, we provide a theoretical derivation based on the generator's mechanics. Let $G(\cdot; \Theta)$ be the frozen generator parameterized by $\Theta$. For any class $c$, the feature of

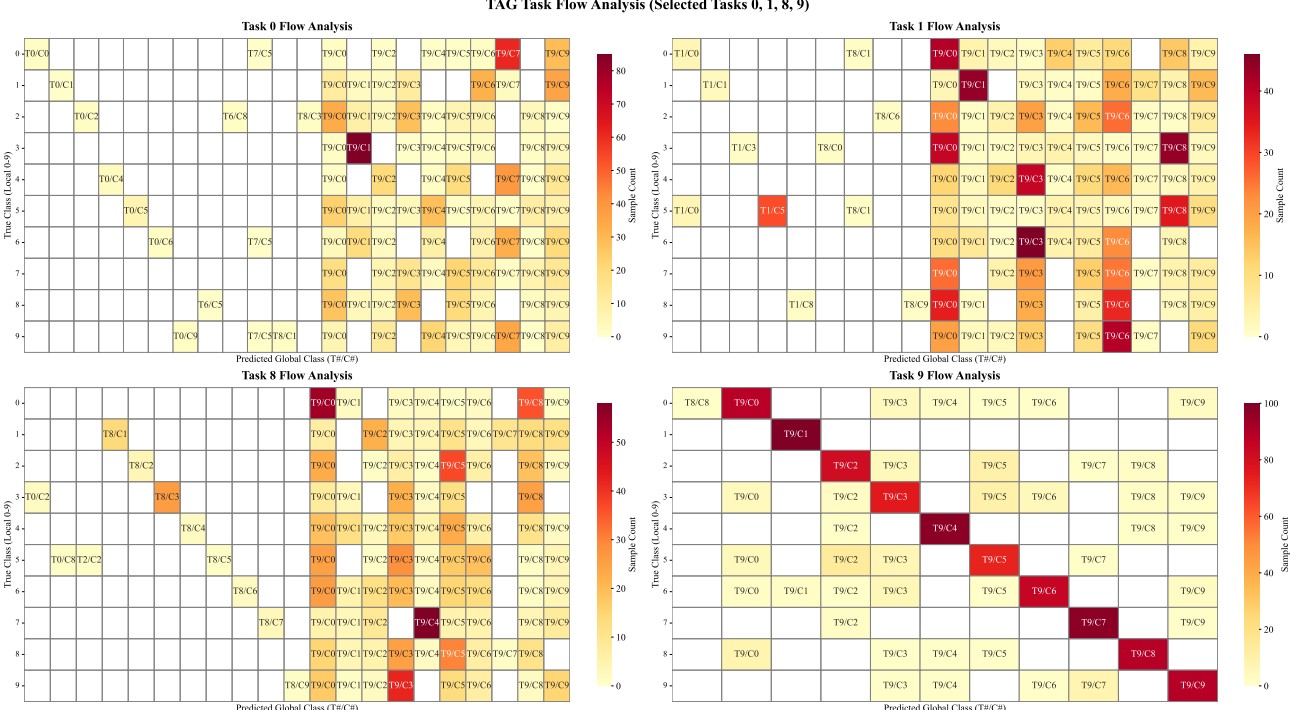

*Figure 8.* **Empirical Verification of Domain Shortcut**. We visualize the Task-Agnostic Accuracy (TAG) classification flow for Task 0/1/8/9 samples (Real Data) on the CIFAR-100 dataset. The x-axis represents the predicted global class index. "T$i$/C$j$" means the $j$th Class of the $i$th Task. Consistent with our theoretical derivation, the heatmap reveals a massive concentration of predictions in the Task 9 region (T9/C#, highlighted in red), which corresponds to the current Real task. This empirical evidence confirms that the model correctly identifies the domain attribute ("Real") but incorrectly associates it with the only available Real classes (Task 9), verifying the existence of the "Domain Shortcut" where domain cues override semantic distinctions.

generated sample $z_{syn}$ can be modeled as:

$$z_{syn}^c = G(\text{prompt}_c; \Theta) \approx z_{ideal}^c + \mathcal{A}(\Theta) + \epsilon, \tag{26}$$

where $z_{ideal}^c$ denotes the perfect semantic representation, and $\mathcal{A}(\Theta)$ represents the systematic artifacts (*e.g.*, checkerboard patterns, spectral bias) inherent to the frozen weights $\Theta$. Crucially, in our Training-Free Generative Replay setting, the generator parameters $\Theta$ are **shared and frozen** across all incremental stages and all classes. Consequently, the artifact term $\mathcal{A}(\Theta)$ is structurally consistent and relatively invariant to the conditioning class $c$. In contrast, real images $z_{real}^c$ are free from these specific generative artifacts: $z_{real}^c \approx z_{ideal}^c + \epsilon$.

Subtracting the expectations of real and synthetic features yields the domain bias vector for class $c$:

$$\mathbb{E}[z|c, real] - \mathbb{E}[z|c, syn] \approx (z_{sem}(c)) - (z_{sem}(c) + \mathcal{A}(\Theta)) = -\mathcal{A}(\Theta). \tag{27}$$

Since the term $-\mathcal{A}(\Theta)$ is governed by the frozen generator and is independent of the specific class label $c$, the domain bias vectors for all classes lie within the same low-dimensional manifold spanned by these systematic artifacts. Let this manifold be the subspace $\mathcal{S}_{dom}$. Therefore, we conclude that the domain shift is shared across all tasks:

$$\forall c \in \mathcal{C}_{1:T}, \quad (\mathbb{E}[z|c, real] - \mathbb{E}[z|c, syn]) \in \mathcal{S}_{dom}. \tag{28}$$

This justifies using the subspace estimated from new classes to rectify old classes.

### A.2.2. EMPIRICAL VERIFICATION: CROSS-CLASS GENERALIZATION

To empirically validate that $\mathcal{S}_{dom}$ is indeed shared, we conduct a **Cross-Class Generalization Experiment** on CIFAR-100. We estimate the domain projection matrix $P_{new}$ using *only* the data from New Classes (Task $T$). We then apply this projector

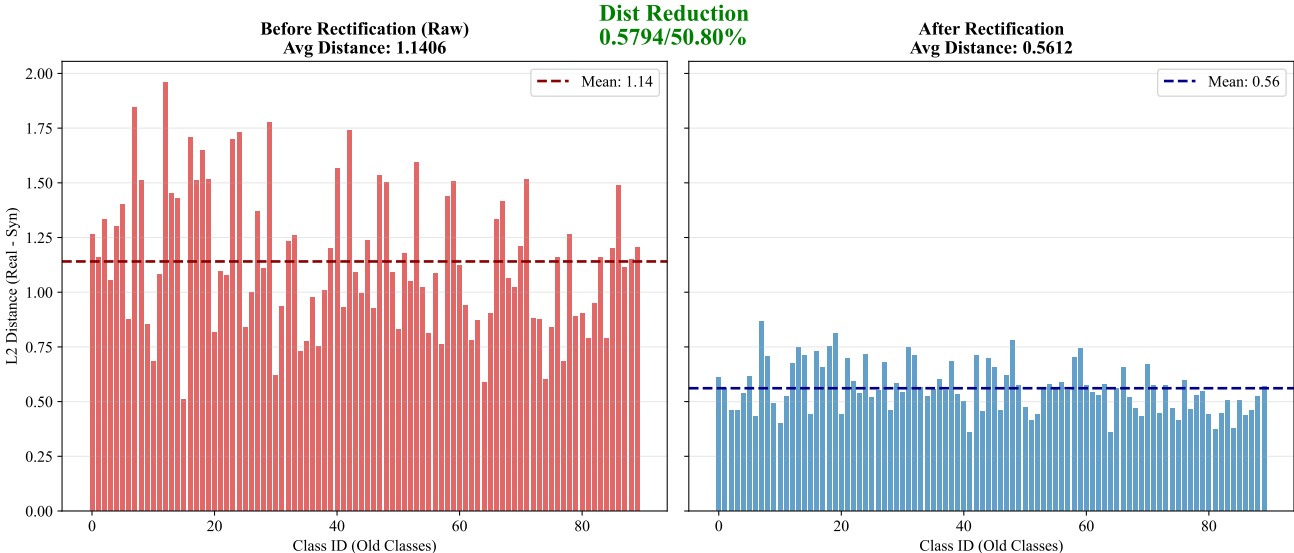

*Figure 9.* **Empirical Verification of Shared Domain Subspace (Corollary 3.2).** We evaluate whether the domain subspace estimated from **New Classes** can generalize to **Old Classes**. The histograms show the $L_2$ distance between real and synthetic feature centroids for each unseen old class. **Left:** Raw features exhibit a significant domain gap (Avg Dist: 1.14). **Right:** After applying the rectification matrix $P$ learned *exclusively* from new classes, the domain gap for old classes is reduced by **50.80%** (Avg Dist: 0.56). This successful cross-class generalization confirms that the domain bias lies in a shared subspace across disjoint categories.

to the data of unseen Old Classes (Task $T - 1$) to see if it reduces their domain gap. We measure the Maximum Mean Discrepancy (MMD) between Real and Synthetic distributions for Old Classes before and after projection.

As illustrated in Figure 9, the projector learned from disjoint new classes successfully aligns the distributions of old classes, reducing the average domain distance from **1.14 to 0.56 (a 50.80% reduction)**, which empirically proves that $\Delta_{dom}$ resides in a shared subspace $\mathcal{S}_{dom}$. This finding is further corroborated qualitatively by the t-SNE visualization in Figure 7, where the rectified features for Task 0 (Old Classes) are transformed using the projection matrix derived *exclusively* from New Classes. The fact that these features transition from dispersed raw distributions to compact, well-clustered representations strongly validates that the domain artifacts identified in new tasks are consistent with those in old tasks, confirming the cross-class generalizability of our rectification strategy.

The above analysis and experimental results validate Corollary 3.2.

### A.3. Analysis and Verification of Corollary 3.3

**Corollary 3.3 Restated.** *Since synthetic data is produced by a training-free generator exhibiting specific stylistic tendencies and patterns, we suppose that the domain bias is not random noise but stems from inherent systematic generative artifacts. This bias manifests as a **structured component** $\Sigma_{dom}$ aligned with specific directions in the feature space. In contrast, we model intra-class semantic variations (e.g., color, pose, background) as **statistically isotropic components** $\sigma_{sem}^2 I$. Mathematically, the covariance of class-conditional residuals satisfies:*

$$\mathrm{Cov}(z|c) \approx \Sigma_{dom} + \sigma_{sem}^2 I, \quad \text{s.t. } \lambda_1(\Sigma_{dom}) \gg \sigma_{sem}^2. \tag{29}$$

#### A.3.1. MATHEMATICAL DERIVATION VIA RESIDUAL COVARIANCE

We rigorously derive the approximation $\mathrm{Cov}(z|c) \approx \Sigma_{dom} + \sigma_{sem}^2 I$ through the following steps.

**Step 1: Define the Residual Vector** $r$ In our method, we compute the residual of a real sample relative to the synthetic center. For a sample $i$ of class $c$, the residual is defined as:

$$r_i = z_i^{real} - \mu_c^{syn}. \tag{30}$$

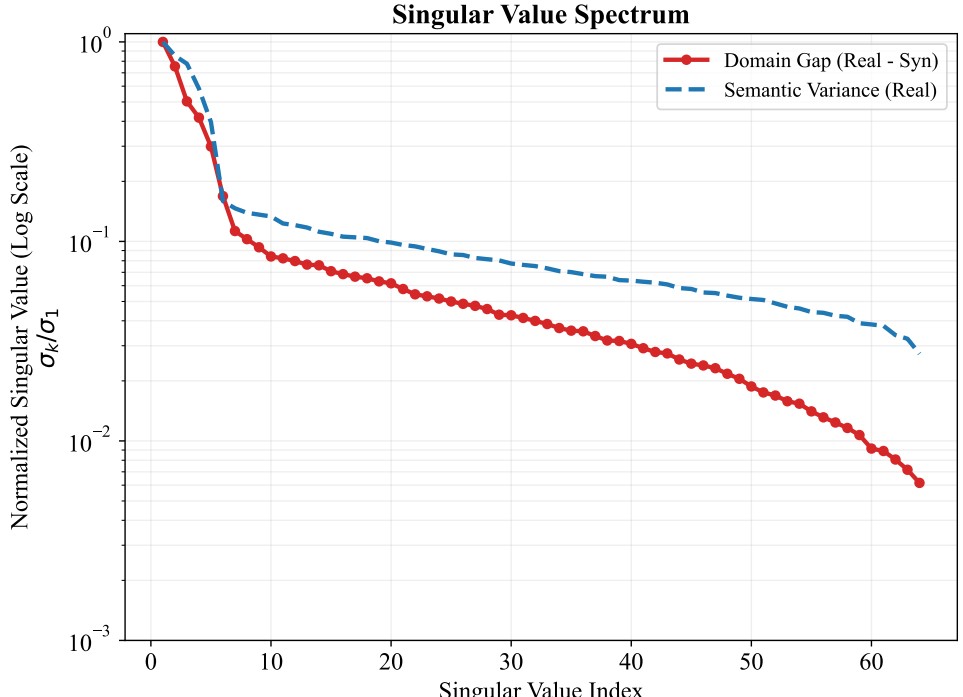

*Figure 10.* **Empirical Verification of Corollary 3.3 (Singular Value Spectrum).** We analyze the rank structure of the Domain Gap (Red, Solid) and Intra-class Semantic Variance (Blue, Dashed) using SVD on a logarithmic scale. The **Domain Gap** curve exhibits a precipitous *exponential decay*, confirming that the systematic domain artifacts are concentrated in a low-dimensional subspace (Low-Rank). In contrast, the **Semantic Variance** curve shows a slower *power-law decay* with a heavy tail, indicating that semantic information is distributed robustly across dimensions. The clear spectral crossover at $k \approx 6$ (where Blue > Red) validates our strategy of excising the top-$k$ dominant domain components while preserving the semantic-rich residual subspace.

Using the feature decomposition $z = z_{sem} + z_{dom} + \epsilon$, we expand $r_i$:

$$
\begin{aligned}
r_i &= (z_{sem,i} + z_{dom}^{real}) - (\bar{z}_{sem,c} + z_{dom}^{syn}) \\
&= \underbrace{(z_{dom}^{real} - z_{dom}^{syn})}_{\text{Domain Shift } \Delta_{dom}} + \underbrace{(z_{sem,i} - \bar{z}_{sem,c})}_{\text{Semantic Fluctuation } s_i} + \epsilon_i.
\end{aligned}
\tag{31}
$$

Here, $\Delta_{dom}$ represents the constant domain shift vector (since the generator is frozen and shared), and $s_i$ represents the zero-mean semantic fluctuation of the sample relative to its class mean.

**Step 2: Compute the Covariance Matrix** The covariance matrix (specifically, the second moment matrix) is defined as the expectation $\mathbb{E}[rr^T]$.

$$
\begin{aligned}
\text{Cov}(r) &= \mathbb{E}[r_i r_i^T] \\
&= \mathbb{E}[(\Delta_{dom} + s_i)(\Delta_{dom} + s_i)^T] \\
&= \mathbb{E}[\Delta_{dom}\Delta_{dom}^T + s_i s_i^T + \Delta_{dom}s_i^T + s_i\Delta_{dom}^T].
\end{aligned}
\tag{32}
$$

**Step 3: Apply Statistical Corollary** We analyze each term in the expansion:

- **Term 1:** $\Delta_{dom}\Delta_{dom}^T$. This is the outer product of a constant vector. Linear algebra dictates that its rank is exactly 1. This corresponds to the perfectly structured, low-rank component $\Sigma_{dom}$.

- **Term 2:** $\mathbb{E}[s_i s_i^T]$. This represents the covariance of intra-class semantic variations. Relative to the dominant domain shift, semantic variations (*e.g.*, pose, background) are diverse and high-dimensional. For mathematical tractability, we model this as isotropic noise: $\mathbb{E}[s_i s_i^T] \approx \sigma_{sem}^2 I$.

- **Terms 3 & 4: Cross-Terms** $\mathbb{E}[\Delta_{dom}s_i^T]$. Since $\Delta_{dom}$ is constant and $s_i$ is a zero-mean random variable, and assuming domain artifacts are uncorrelated with specific semantic contents (*e.g.*, image quality is independent of object pose), the expectation of cross-terms is 0.

**Step 4: Final Formulation** Combining these terms, we arrive at the final approximation:

$$\text{Cov}(r) \approx \underbrace{\Delta_{dom}\Delta_{dom}^T}_{\Sigma_{dom} \text{ (Low-Rank)}} + \underbrace{\sigma_{sem}^2 I}_{\text{Isotropic Noise}} . \tag{33}$$

### A.3.2. EMPIRICAL VERIFICATION: SINGULAR VALUE SPECTRUM

To empirically validate Corollary 3.3 regarding the rank structure of domain and semantic components, we conduct a singular value spectrum analysis. We construct the domain difference matrix according to Hierarchical Subspace Rectification described in main manuscript. Besides, we construct the semantic matrix $M_{sem}$ by concatenating class-centered real features from all categories. SVD is applied to estimate the rank structure of intra-class semantic variations. To highlight the rate of energy decay, we plot the normalized singular values ($\sigma_k/\sigma_1$) on a logarithmic scale. We analyze the singular value spectrum on CIFAR-100 ($T = 10$), as shown in Figure 10.

First, observing the **Domain Gap (Red Curve)**, we identify a sharp *exponential decay* in the logarithmic scale. The energy is heavily concentrated in the first few principal components ($k \leq 5$), after which the singular values drop precipitously below $10^{-2}$. This empirically confirms that the systematic artifacts introduced by the frozen generator are confined to a very low-dimensional subspace $\mathcal{S}_{dom}$.

Second, in contrast, the **Semantic Variance (Blue Curve)** exhibits a much slower *power-law decay*. While not perfectly isotropic (flat) due to the manifold nature of deep features, it maintains a "heavy tail", indicating that discriminative semantic information is distributed across a wide range of feature dimensions.

**The Crucial Insight lies in the Spectral Disparity:** In the dominant components (Indices 1-5), the domain bias overwhelms the semantic signal, which explains the prevalence of the Domain Shortcut. However, a crossover occurs rapidly around $k \approx 5$. Beyond this point, the semantic variance consistently dominates the residual domain noise (*i.e.*, Semantic $\gg$ Domain). This spectral separation provides robust justification for our Feature Domain Rectification (FDR) strategy: by excising only the top-$k$ domain directions, we effectively eliminate the major source of domain shift while preserving the semantic-rich residual subspace intact.

### A.4. Proof of Soft Projection Necessity via SNR Analysis

We quantify the effectiveness of FDR using the Signal-to-Noise Ratio (SNR). Let "Signal" be semantic variance and "Noise" be domain variance. Before rectification, due to the Domain Shortcut, $\text{Tr}(\Sigma_{dom}) \gg \text{Tr}(\Sigma_{sem})$, implying $\text{SNR}_{raw} \to 0$.

The FDR projection matrix is $P = I - \gamma U_{dom}U_{dom}^T$. After projection, the domain variance is suppressed by $(1-\gamma)^2$, while semantic variance is preserved (assuming near-orthogonality).

$$\text{SNR}_{rect} \approx \frac{\text{Tr}(P\Sigma_{sem}P^T)}{\text{Tr}(P\Sigma_{dom}P^T)} \approx \frac{\text{Tr}(\Sigma_{sem})}{(1-\gamma)^2\text{Tr}(\Sigma_{dom})}. \tag{34}$$

- If $\gamma = 1.0$ (Hard Projection), the denominator becomes 0, theoretically maximizing SNR. However, due to feature entanglement, this also cuts into the numerator (Signal), causing performance drops.

- If $\gamma \approx 0.8$ (Soft Projection), the domain noise is attenuated by a factor $(1 - 0.8)^2 = 0.04$. The noise is reduced by 96%, making it negligible, while the signal remains intact. This maximizes the *effective* SNR for classification.

## B. Methodological Discussions

In this section, we provide a rigorous justification for the design choices in the Feature Domain Rectification (FDR) framework, specifically focusing on the residual definition (Eq. 5) and the dual-stage logic of projection matrix calculation versus application.

## B.1. Justification for Residual Definition (Eq. 5)

We define the residual for a real sample $z_i^{real}$ of class $c$ as $r_i = z_i^{real} - \mu_c^{syn}$. To demonstrate the necessity of this specific formulation, we analyze all plausible alternatives for calculating the difference between the Real and Synthetic domains:

1. **Case 1: Sample-to-Sample** ($z_i^{real} - z_j^{syn}$)
   *Drawback: High Variance from Generative Noise.*
   Synthetic samples often exhibit stochastic instability (*e.g.*, mode collapse or high-frequency artifacts). Computing residuals against random synthetic instances introduces significant instance-level noise $\epsilon_{syn}$. This increases the variance of the residual estimates, making it difficult to isolate the consistent domain bias direction.

2. **Case 2: Center-to-Center** ($\mu_c^{real} - \mu_c^{syn}$)
   *Drawback: Loss of Instance Granularity.*
   While the mean difference vector captures the global domain shift, it assumes a uniform bias magnitude for all samples. However, real samples are distributed diversely in the feature space. A simple center-to-center subtraction fails to capture how specific instances deviate from the domain boundary, treating the diverse semantic manifold as a single point.

3. **Case 3: Synthetic-Sample-to-Real-Center** ($z_j^{syn} - \mu_c^{real}$)
   *Drawback: Misaligned Reference Frame.*
   In the context of generative replay, our goal is to rectify *real* samples to match the *synthetic* replay buffer because old classes only have synthetic data, not vice versa. Using $\mu_c^{real}$ as the anchor would model the inverse domain shift (Synthetic → Real), which is irrelevant to the inference-time task where only real samples are available.

4. **Ours: Real-Sample-to-Synthetic-Center** ($z_i^{real} - \mu_c^{syn}$)
   *Advantage: Stability and Specificity.*
   We use the synthetic class centroid $\mu_c^{syn}$ as a **stable anchor** (averaging out generative noise) and the real instance $z_i^{real}$ as the **source**. This formulation captures the specific domain deviation of each real sample relative to the ideal target distribution.

## B.2. Decoupling Subspace Estimation from Instance Rectification

A critical methodological question is: *Why do we calculate the projection matrix $P$ on uncentered features, but apply it to centered residuals?* This apparent contradiction stems from the distinct roles of the two phases: **Estimation** vs. **Rectification**.

### B.2.1. PHASE 1: THE ESTIMATION PHASE

**Objective:** To identify the dominant enemy—the Domain Shift. The domain gap mathematically consists of two components:

- **First-Order Bias (Mean Shift):** The distance between distribution centers ($\mu_{real} - \mu_{syn}$).

- **Second-Order Bias (Covariance Shift):** The difference in distribution shapes.

**Mathematical Derivation:** Let the uncentered difference vector be $\text{Diff} = z_{real} - \mu_{syn}$. We can decompose it as:

$$\text{Diff} = \underbrace{(z_{real} - \mu_{real})}_{\text{A: Semantic Variance}} + \underbrace{(\mu_{real} - \mu_{syn})}_{\text{B: Domain Mean Shift}} . \tag{35}$$

- **Term A (Variance):** Represents diverse semantic information (*e.g.*, object pose), which directions are scattered.

- **Term B (Bias):** Represents the consistent domain shift, which has a constant direction and high magnitude.

**Why Uncentered SVD?** When we perform SVD on the uncentered Diff matrix: Since Term B is consistent across all samples and possesses high energy, the top singular vector $v_1$ (where $k = 1$) will align precisely with the direction of the Mean Shift ($\mu_{real} - \mu_{syn}$).

**Counterfactual:** If we were to center the data first ($z'_{real} = z_{real} - \mu_{real}$), Term B would vanish (**0**). SVD would then only capture Term A (semantic shape differences). Cutting these directions would damage semantic discriminability (*e.g.*, making the model unable to distinguish a standing dog from a sitting one).

*Conclusion: We must NOT center during estimation to preserve the "Mean Shift" signal for SVD detection.*

### B.2.2. PHASE 2: THE RECTIFICATION PHASE

**Objective:** To eliminate the identified bias (aligning the blocks). The rectification process $\tilde{z} = z - PP^T(z - \mu_{syn})$ operates in two steps:

**Step 1: Centering (Translation Alignment)**

$$z' = z - \mu_{syn} \tag{36}$$

This step addresses the **First-Order Bias**. It translates the real data manifold so that its reference point aligns with the synthetic origin. Without this step, the two distributions remain far apart in Euclidean space, making subsequent rotation/projection ineffective.

**Step 2: Projection (Geometric Rectification)**

$$\tilde{z} \leftarrow Pz' \tag{37}$$

This step addresses the **structural consistency**. Since $P$ captures the direction of the domain artifact (identified in Phase 1), this projection excises the component of the vector that aligns with the domain subspace. *Crucially, projection matrices operate on vector directions relative to the origin.* By applying $P$ to the centered residual, we ensure that we are removing the "shape/direction" anomaly relative to the target anchor, rather than distorting the absolute position of the features.

**Summary:** We use uncentered features for estimation to **detect** the global mean shift, and we use centered residuals for application to **correct** the relative geometric deviation.

## C. Algorithms and Implementation Details

### C.1. Details of Attribute-Aware Generative Replay

In this section, we provide a comprehensive description of our data generation pipeline, corresponding to Algorithm 1.

**Attribute-Aware Prompt Extraction.** Instead of relying on generic class names (*e.g.*, "a photo of a dog"), which often lead to ambiguous or stereotypical generations, we leverage a Vision-Language Model (VLM) to extract instance-specific semantic details. Specifically, we employ Qwen2.5-VL to interrogate each real image in the current training stream. For every image, we construct a structured query focusing on five aspect-oriented attributes:

- "What is the fine-grained category of this {label}?"

- "What is the color of this {label}?"

- "What is the pose of this {label}?"

- "What is the background of this image?"

- "What is the {label} doing in this image?"

The answers to these questions are concatenated to form a dense, high-fidelity semantic caption. These captions are stored in a Prompt Bank, which incurs negligible storage overhead compared to saving raw pixel data.

**Image Synthesis with Training-Free Generator.** At the beginning of a new task $t$, we retrieve the stored captions for all previous classes $(1 \ldots t)$ from the Prompt Bank. These rich text descriptions are fed into a frozen pre-trained Text-to-Image (T2I) generator (we utilize Qwen-Image in our experiments) to synthesize the raw replay dataset. This approach ensures that the generated images cover diverse attributes (*e.g.*, various poses and backgrounds) historically recorded from the real data distribution.

---

**Algorithm 1** Attribute-Aware Generative Replay with Task-Constraint Filtering

---

**Require:** Current task training data $\mathcal{D}_t$, Old model $M_{t-1}$ (if $t > 0$), Vision-Language Model $\mathcal{V}$, Text-to-Image Generator $\mathcal{G}$, Prompt Bank $\mathcal{B}$.

**Ensure:** Synthetic Replay Dataset $\mathcal{D}_{syn}$.

1: **Stage 1: Attribute-Aware Prompt Extraction**
2: **for** each real image $(x, y)$ in $\mathcal{D}_t$ **do**
3:     Define question set $Q = \{category, color, pose, background, action\}$
4:     Extract dense caption $p \leftarrow \mathcal{V}(x, Q)$ {Concatenate answers}
5:     Update Prompt Bank: $\mathcal{B} \leftarrow \mathcal{B} \cup \{(p, y)\}$
6: **end for**
7: **Stage 2: Synthesis and Filtering**
8: Initialize $\mathcal{D}_{syn} \leftarrow \emptyset$
9: **for** task $k = 0$ to $t$ **do**
10:     Retrieve prompts $\mathcal{P}_k$ for task $k$ from $\mathcal{B}$
11:     **for** each $(p, y) \in \mathcal{P}_k$ **do**
12:         Generate candidate image: $\hat{x} \leftarrow \mathcal{G}(p)$
13:         **if** $k < t$ **and** $M_{t-1}$ exists **then**
14:             // **Case A: Old Task Replay** $\rightarrow$ **Apply Filtering**
15:             Get class set $\mathcal{C}_k$ for task $k$
16:             $\hat{y} = \operatorname{argmax}_{c \in \mathcal{C}_k} P(c|\hat{x}; \theta_{t-1})$
17:             **if** $\hat{y} == y$ **then**
18:                 $\mathcal{D}_{syn} \leftarrow \mathcal{D}_{syn} \cup \{(\hat{x}, y)\}$
19:             **end if**
20:         **else**
21:             // **Case B: New Task Augmentation** $\rightarrow$ **No Filtering**
22:             {Old model cannot filter classes unknown to it}
23:             $\mathcal{D}_{syn} \leftarrow \mathcal{D}_{syn} \cup \{(\hat{x}, y)\}$
24:         **end if**
25:     **end for**
26: **end for**
27: **Return** $\mathcal{D}_{syn}$

---

**Task-Constraint Data Filtering.** To prevent the generated images from polluting the decision boundary, we implement a filtering strategy using the frozen old model $M_{t-1}$. A naive global classification filtering is prone to inter-task confusion. Instead, we propose a Task-Constraint filtering mechanism. For a synthetic sample $x_{syn}$ conditioned on label $y$ belonging to an old task $k$, we utilize the task identity as a prior. We retain the sample only if the old model correctly classifies it within the output space of its specific task $\mathcal{C}_k$:

$$\operatorname{argmax}_{c \in \mathcal{C}_k} P(c|x_{syn}; \theta_{t-1}) = y \tag{38}$$

This constraint ensures that the synthetic old-class data maintains high semantic consistency with the intra-task decision boundaries established by the old model, effectively filtering out ambiguous samples while avoiding interference from classes in other tasks and storing them as $\mathcal{D}_{old}^{syn}$. For synthetic new-class data, we directly retain them as $\mathcal{D}_{new}^{syn}$

### C.2. DREAM Training Procedure

In this section, we detail the complete training loop in Algorithm 2 and how the Rectification Matrix $P$, Domain Means $\mu$, and Class Prototypes $\mathcal{C}$ are updated, focusing on their frequency and the caching mechanism.

#### C.2.1. EPOCH-WISE STATISTICAL UPDATE STRATEGY

To balance the adaptability of the feature space with computational efficiency, we decouple the update frequency of the model parameters from the rectification statistics.

- **Model Weights** $\theta$: Updated at every iteration step via stochastic gradient descent (Step-wise).

---

**Algorithm 2** DREAM Training Loop

---

**Require:** Current model $\mathcal{M}_t$, Frozen old model $\mathcal{M}_{t-1}$ (with old stats $P_{t-1}, \mu_{t-1}$), Current Data $\mathcal{D}_t^{real}$, Synthetic Old $\mathcal{D}_{old}^{syn}$, Synthetic New $\mathcal{D}_{new}^{syn}$, Cache $\mathcal{H}$.
 1: **Hyperparams:** $\gamma = 0.8, \lambda_{KD} = 1.0, \lambda_{Proto} = 0.1, k = 5$
 2: Initialize Model $\mathcal{M}_t \leftarrow \mathcal{M}_{t-1}$, Feature Cache $\mathcal{H} \leftarrow \emptyset$, Prototypes $\mathcal{C} \leftarrow \emptyset$
 3: **for** epoch = 1 to $E$ **do**
 4:     // **Step 1: Update FDR Stats & Prototypes (Efficiently)**
 5:     **if** epoch == 1 **then**
 6:         Extract features $Z, Y, D$ from subset of $\mathcal{D}_{all}$ {Initial calculation}
 7:     **else**
 8:         Retrieve $Z, Y, D$ from $\mathcal{H}$ {Use cached features}
 9:     **end if**
10:     Compute residuals $R = Z - \mu_{syn}[Y]$ based on domain $D$
11:     Perform Hierarchical SVD on $R$ to obtain $U_{dom}$ {top5 vector}
12:     Construct Projection Matrix $P = I - \gamma U_{dom} U_{dom}^T$
13:     Update Domain Means $\mu_{real}, \mu_{syn}$
14:     **Update Prototypes:**
15:     Calculate rectified features $\tilde{Z} = P(Z - (\mathbb{I}_{D=real}\mu_{real} + \mathbb{I}_{D=syn}\mu_{syn}))$
16:     Update Prototypes $\mathcal{C}_c \leftarrow \text{Mean}(\tilde{Z}[Y = c])$ for each class $c$
17:     Reset Cache $\mathcal{H} \leftarrow \emptyset$
18:     // **Step 2: Training & Caching**
19:     **for** batch $(x, y, d)$ in Dataloader **do**
20:         **Current Forward:** $z = \mathcal{M}_t(x)$
21:         **Cache Features:** $\mathcal{H}$.append($z$.detach(), $y, d$)
22:         **Teacher Forward:**
23:         $z_{old\_rect} = P_{t-1}(\mathcal{M}_{t-1}(x) - \mu_{t-1})$ {Rectify using old stats}
24:         **Rectification (Current):**
25:         $z' = z - (\mathbb{I}_{d=real}\mu_{real} + \mathbb{I}_{d=syn}\mu_{syn})$
26:         $\tilde{z} = Pz'$
27:         Compute Logits $L = \text{Classifier}(\tilde{z})$
28:         $\mathcal{L}_{CE} = \text{CrossEntropy}(L, y)$
29:         $\mathcal{L}_{Proto} = \text{RAPC}(\tilde{z}, \mathcal{C})$ {Using updated Prototypes}
30:         $\mathcal{L}_{KD} = \text{Distillation}(\tilde{z}, z_{old\_rect})$ {Using old rectified features}
31:         Update weights via $\nabla(\mathcal{L}_{CE} + \lambda_{Proto}\mathcal{L}_{Proto} + \lambda_{KD}\mathcal{L}_{KD})$
32:     **end for**
33: **end for**

---

- **Rectification Statistics** $(P, \mu)$**:** Updated once per epoch (Epoch-wise).

- **Class Prototypes** $\mathcal{C}$**:** Updated once per epoch, immediately following the rectification update.

This epoch-wise strategy assumes that the feature distribution shifts slowly within a single epoch, allowing us to treat the geometric manifold as locally stationary.

C.2.2. EFFICIENT CACHING MECHANISM

Calculating the SVD for the projection matrix $P$ requires a global view of the feature distribution. A naive implementation would require an additional forward pass over the dataset at the beginning of each epoch, effectively doubling the computational cost. To address this, we introduce a **Ping-Pong Caching Mechanism** as described in Algorithm 2:

1. **Initialization (Epoch 1):** Since no cache exists, we perform an explicit inference pass on a subset of data to calculate the initial $P$ and $\mu$.

2. **Online Caching (Epoch $t \rightarrow t + 1$):** During the training iterations of Epoch $t$, while performing the standard forward

*Table 7.* Detailed Hyperparameters and Implementation Settings.

| Configuration | CIFAR10/CIFAR100/TinyImageNet | ImageNet-Subset |
|---|---|---|
| *Optimization* | | |
| Backbone | ResNet-32 | ResNet-18 |
| Optimizer | SGD (Momentum=0.9) | SGD (Momentum=0.9) |
| Learning Rate | 0.1 (Decay $5 \times 10^{-4}$ at milestones) | 0.1 (Decay $5 \times 10^{-4}$ at milestones) |
| Batch Size | 128 | 128 |
| Epochs | 200/100 | 40/40 |
| *Generative Replay* | | |
| Generator | Qwen-Image | Qwen-Image |
| VLM Attribute Extractor | Qwen2.5-VL | Qwen2.5-VL |
| Images per Class ($N_{syn}$) | 5000/500/500 | 1300 |
| *DREAM Specifics* | | |
| Rectification Strength $\gamma$ | 0.8 | 0.8 |
| Subspace Dimension $k$ | 5 | 5 |
| $\lambda_{Proto}$ | 0.1 | 0.1 |
| $\lambda_{KD}$ | 1.0 | 1.0 |

pass for loss computation, we detach and store the extracted features $z$ into a temporary cache $\mathcal{H}$.

3. **Zero-Cost Retrieval:** At the start of Epoch $t + 1$, we directly retrieve the features from $\mathcal{H}$ to compute the new $P$ and $\mu$. This eliminates the need for a separate extraction pass for all subsequent epochs.

### C.2.3. UPDATE PROTOCOLS

**1. Domain Statistics and Projection Matrix.** At the start of each epoch, utilizing the cached features $Z \in \mathbb{R}^{N \times D}$, labels $Y$, and domain tags $D$, we first compute the domain-specific means $\mu_{real}$ and $\mu_{syn}$. The domain gap residuals are computed as:

$$R_i = z_i - \mu_{syn}[y_i], \quad \text{where } d_i = \text{real} \tag{39}$$

We then apply SVD on the stacked residual matrix $R$ to obtain the principal directions of the domain gap $U_{dom}$. The projection matrix is updated as $P = I - \gamma U_{dom} U_{dom}^T$.

**2. Prototype Refreshment.** Crucially, the class prototypes must remain aligned with the current rectified feature space. Immediately after updating $P$ and $\mu$, we re-rectify the cached features $Z$ using the *new* statistics:

$$z_i' = P_{new}(z_i - \mu_{new}[d_i]) \tag{40}$$

The prototype for class $c$ is then refreshed by averaging these newly rectified features:

$$\mathcal{C}_c = \frac{1}{|\{i : y_i = c\}|} \sum_{i:y_i=c} z_i' \tag{41}$$

This ensures that the Prototype Loss pushes samples towards the current, geometrically corrected class centers.

**3. Frozen Teacher Rectification.** For the knowledge distillation term $\mathcal{L}_{KD}$, we employ the old model $\mathcal{M}_{t-1}$ as the teacher. It is important to note that the teacher's output must also be rectified to provide consistent supervision. We utilize the **frozen statistics** $(P_{t-1}, \mu_{t-1})$ saved from the end of the previous task to rectify the teacher's features. This guarantees that the student learns to match the teacher's "clean" feature representation rather than its raw, biased output.

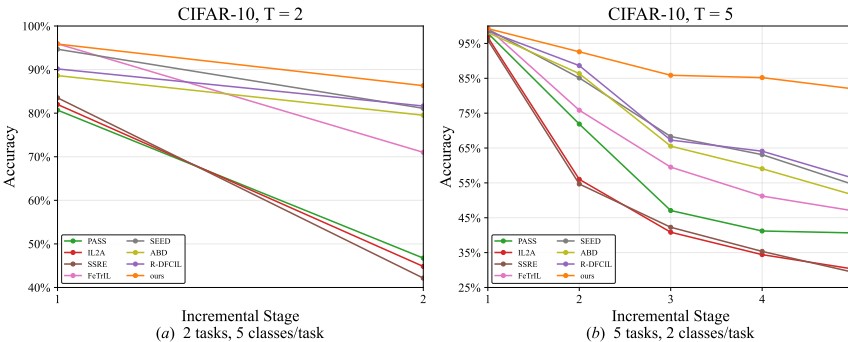

*Figure 11.* Task-agnostic average accuracy curves on CIFAR-10. Results for (a) $T = 2$ tasks and (b) $T = 5$ tasks show that DREAM (orange curve) consistently outperforms state-of-the-art exemplar-free methods.

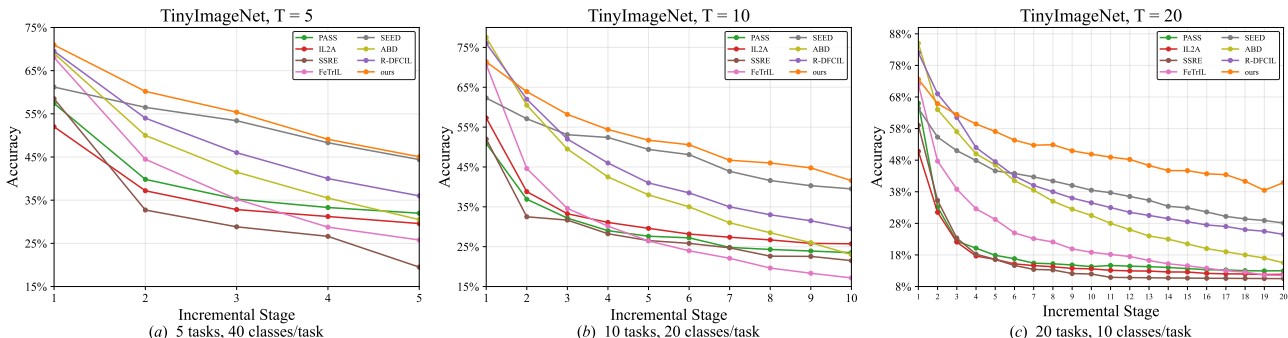

*Figure 12.* Task-agnostic average accuracy curves on TinyImageNet. Results for (a) $T = 5$ tasks, (b) $T = 10$ tasks, and (c) $T = 20$ tasks show that DREAM (orange curve) consistently outperforms state-of-the-art exemplar-free methods.

### C.3. Hyperparameters

In this section, we provide a comprehensive overview of the hyperparameters and implementation details used in our experiments, listed in Table 7. Our framework is implemented using PyTorch (Paszke et al., 2019), and all experiments are conducted on standard benchmarks including CIFAR-10, CIFAR-100, TinyImageNet, and ImageNet-Subset.

**Architecture and Optimization.** For the backbone network, we employ a **ResNet-32** for the CIFAR-10, CIFAR-100, and TinyImageNet datasets, while a **ResNet-18** is utilized for the ImageNet-Subset. Across all datasets, the models are optimized using Stochastic Gradient Descent (SGD) with a momentum of 0.9. We set the initial learning rate to 0.1, which decays at specific milestones during training. The weight decay is set to $5 \times 10^{-4}$. We use a consistent batch size of 128 for all experiments. Regarding the training duration, models are trained for 200 epochs for initial task 0 and 100 epochs for incremental task 1-9 on CIFAR-10/100 and TinyImageNet, whereas for the larger ImageNet-Subset, the training is restricted to 40 epochs per task to maintain efficiency.

**Generative Replay Configuration.** Our generative replay mechanism relies on **Qwen2.5-VL** as the attribute-aware prompt extractor and **Qwen-Image** as the training-free generator. To balance memory costs and performance, the number of synthetic samples generated per class ($N_{syn}$) varies by dataset complexity: we generate 5,000 images per class for CIFAR-10, 500 images per class for both CIFAR-100 and TinyImageNet, and 1,300 images per class for ImageNet-Subset.

**DREAM Hyperparameters.** Our proposed method introduces four key hyperparameters, which are kept consistent across all datasets to demonstrate robustness. Specifically, the rectification strength is set to $\gamma = 0.8$, and the subspace dimension for the SVD projection is set to $k = 5$. For the loss functions, the weight for the prototype loss is set to $\lambda_{Proto} = 0.1$, and the knowledge distillation weight is set to $\lambda_{KD} = 1.0$.

*Table 8.* **Quantitative Analysis of the "Apples-to-Apples" Baseline.** We compare our full DREAM framework against *Raw Generative Replay*, which utilizes the same synthesis data but excludes the FDR and RAPC module. Despite the visual quality of the synthetic data, the baseline suffers a catastrophic performance drop. This empirically confirms that raw synthetic features introduce severe domain shortcuts, and our DREAM mechanism is essential for aligning the synthetic feature distribution with real data.

| Methods | CIFAR-10 | | CIFAR-100 | | | ImageNet-Subset | | | TinyImageNet | | |
|---|---|---|---|---|---|---|---|---|---|---|---|
| | T=2 | T=5 | T=5 | T=10 | T=20 | T=5 | T=10 | T=20 | T=5 | T=10 | T=20 |
| Raw Generative Replay | 76.79 | 55.48 | 44.85 | 33.27 | 24.38 | 58.86 | 47.41 | 39.73 | 36.69 | 24.68 | 18.12 |
| **DREAM (Ours)** | **91.08** | **88.98** | **71.55** | **69.73** | **68.48** | **78.37** | **76.57** | **74.43** | **55.54** | **52.92** | **51.00** |
| Δ | +14.29 | +33.50 | +36.70 | +36.46 | +44.10 | +19.51 | +29.16 | +34.70 | +18.85 | +28.24 | +32.88 |

*Table 9.* **Comparison of Task-Aware Accuracy (TAW, %).** We contrast the raw Generative Replay with our Rectified method, DREAM. The high TAW of the baseline confirms effective knowledge retention, while our method further refines discriminability.

| Methods | CIFAR-10 | | CIFAR-100 | | | ImageNet-Subset | | | TinyImageNet | | |
|---|---|---|---|---|---|---|---|---|---|---|---|
| | T=2 | T=5 | T=5 | T=10 | T=20 | T=5 | T=10 | T=20 | T=5 | T=10 | T=20 |
| Raw Generative Replay | 90.63 | 92.90 | 77.70 | 77.16 | 76.53 | 84.79 | 85.96 | 82.31 | 58.84 | 53.74 | 56.92 |
| **DREAM (Ours)** | **95.59** | **96.89** | **83.14** | **86.75** | **91.26** | **85.33** | **87.54** | **86.24** | **64.71** | **67.34** | **71.54** |

# D. Additional Experiments and Analysis

In this section, we provide supplementary experimental results to further validate the robustness, effectiveness, and efficiency of our proposed method.

## D.1. Performance on CIFAR-10 and TinyImageNet

In the main manuscript, we primarily visualized the incremental performance on CIFAR-100 and ImageNet-Subset. To demonstrate the generalization capability of our method across datasets with varying scales and granularities, we provide the Task-Agnostic Accuracy (TAG) curves for **CIFAR-10** and **TinyImageNet** in Figure 11 and Figure 12.

Consistent with the observations in the main manuscript, our method consistently outperforms the state-of-the-art baselines throughout the incremental learning steps. Notably, on TinyImageNet, which presents a higher challenge due to visual complexity, our method maintains a significant lead, verifying that our DREAM is effective regardless of the dataset scale.

## D.2. Raw Generative Replay without FDR and RAPC

To rigorously evaluate the contribution of our proposed Feature Distribution Rectification (FDR) and Real-Anchored Prototype Consolidation (RAPC) mechanism, we construct an "apples-to-apples" baseline denoted as **Raw Generative Replay**. In this setting, we employ the exact same generative model (Qwen-Image) and prompt strategies as our full method to synthesize replay data. However, we strictly remove the FDR and RAPC module, meaning the classifier is trained directly on the raw synthetic features without any distribution alignment or covariance rectification.

Table 8 presents the comparison results on the four benchmarks. We observe that the Raw Generative Replay baseline consistently underperforms compared to our full DREAM framework. This performance drop indicates that relying solely on raw generative replay data makes the model vulnerable to **domain shortcuts**. Instead of learning class-discriminative features, the model tends to overfit to domain-specific attributes to distinguish between real and synthetic data. While the generated images appear visually coherent (as shown in Figure 14), there is a critical distribution shift between the synthetic and real feature spaces. This misalignment in domain features leads the model to over-learn domain cues, causing it to incorrectly classify real old-class data as new, as visualized in **Figure 2 (Main Manuscript)** and **Figure 8 (Appendix)**. This confirms the presence of the domain shortcut. Our FDR module effectively breaks this domain shortcut, preventing the model from exploiting domain cues, leading to a significant improvement in accuracy.

## D.3. Task-Aware Accuracy (TAW) Evaluation

While our main focus is on the more challenging Task-Agnostic Accuracy (TAG) setting, the Task-Aware Accuracy (TAW) metric is crucial for evaluating the discriminability of the learned representation within each task. As mentioned in the introduction, a good generative replay method should maintain high intra-task classification accuracy.

*Table 10.* Comparison with standard domain alignment methods. CORAL and MMD improve over raw replay but remain far below DREAM due to semantic mismatch under disjoint label spaces.

| Method | Raw Replay | CORAL | MMD | DREAM |
|--------|-----------|-------|-----|-------|
| Acc. (%) | 33.27 | 46.34 | 52.81 | **69.73** |

*Table 11.* Effect of real old-class data ratio on CIFAR-100 ($T = 10$). Increasing the amount of real old-class data progressively weakens the synthetic-real asymmetry and improves task-agnostic accuracy.

| Real old-data ratio | 0 | 0.1 | 0.2 | 0.5 | 0.8 | 1.0 |
|--------------------|------|------|------|------|------|------|
| Acc. (%) | 69.73 | 73.21 | 74.63 | 77.78 | **79.97** | 79.37 |

To further understand the failure mode of naive generative replay and the specific contribution of our method, we compare the **TAW** of the Raw Generative Replay without rectification against our proposed DREAM method.

**1. Raw Generative Replay: Strong Anti-Forgetting**

As shown in Table 9, the raw generative replay achieves high TAW performance across all datasets (*e.g.*, 86.75% on CIFAR-100 $T = 10$). Since TAW evaluates classification accuracy within the correct task ID (ignoring inter-task confusion), this high score implies that **Generative Replay inherently possesses strong anti-forgetting capabilities**. The synthetic data, despite its domain artifacts, successfully retains the semantic discriminability required to distinguish classes *within* the same task. The catastrophic drop in Task-Agnostic accuracy (TAG) observed in the main manuscript is **not** due to forgetting semantic knowledge, but solely due to the **"Domain Shortcut"** problem—where the model fails to align disjoint tasks in a shared space. These results further prove the Proposition 3.1.

**2. DREAM: Enhancing Semantic Discriminability**

By applying our Feature Domain Rectification, our method further boosts the TAW performance (*e.g.*, an improvement of **+4.66%** on CIFAR-100 $T = 10$). This result is significant for two reasons:

**Non-Destructive Rectification:** It proves that reducing the domain subspace does not damage the semantic content. If semantic information were coupled with the domain bias, TAW would have dropped.

**Noise Reduction:** The improvement suggests that domain artifacts act as "noise" even for intra-task classification. By filtering out $\Sigma_{dom}$, we produce cleaner feature representations, enabling the classifier to learn sharper decision boundaries.

These results further validate the Corollary 3.2 and Corollary 3.3.

**D.4. Comparison with Standard Domain Alignment Methods**

We further compare DREAM with two standard domain alignment objectives, CORAL and MMD, on CIFAR-100 ($T = 10$). These methods are commonly used to reduce distribution discrepancy between source and target domains. However, they implicitly assume that the aligned domains share overlapping label spaces. This assumption does not hold in exemplar-free generative replay CIL, where the replay domain comprises synthetic samples from the old class, whereas the real domain comprises new-class samples. Directly aligning these two distributions may therefore induce a semantic mismatch.

As shown in Table 10, CORAL and MMD obtain 46.34% and 52.81%, respectively, which are substantially lower than DREAM. This confirms that the main challenge is not merely distribution alignment, but the synthetic-real domain shortcut under disjoint old/new label spaces. DREAM avoids direct global alignment and instead removes the domain-sensitive subspace while preserving semantic structures.

**D.5. Effect of Real Old-Class Data Ratio**

We next analyze when the domain shortcut emerges. Specifically, we gradually introduce real old-class data into the replay data to simulate the transition from strict exemplar-free generative replay to exemplar-based replay or joint training. The real old-class data ratio is varied from 0 to 1, where 0 denotes the strict EFCIL setting with only synthetic old-class replay.

Table 11 shows that increasing the ratio of real old-class data consistently improves performance from 69.73% to 79.97%. This indicates that the domain shortcut is not caused by synthetic data itself, but by the asymmetric structure of strict

*Table 12.* Sensitivity Analysis of Hyperparameters on CIFAR-10. We evaluate the impact of rectification strength $\gamma$, subspace dimension $k$, and weights for prototype loss ($\lambda_{Proto}$) and distillation loss ($\lambda_{KD}$) on model performance. Default settings are bolded.

*(a)* Rectification Strength $\gamma$

| $\gamma$ | Acc. |
|---|---|
| 0.6 | 85.67 |
| 0.7 | 86.97 |
| **0.8** | **88.98** |
| 0.9 | 86.11 |
| 1.0 | 65.78 |

*(b)* Subspace Dimension $k$

| $k$ | Acc. |
|---|---|
| 1 | 70.11 |
| 3 | 85.09 |
| **5** | **88.98** |
| 8 | 86.27 |
| 10 | 79.92 |

*(c)* Proto Weight $\lambda_{Proto}$

| $\lambda_{Proto}$ | Acc. |
|---|---|
| 0 | 86.07 |
| 0.05 | 87.80 |
| **0.10** | **88.98** |
| 0.50 | 84.80 |
| 1.00 | 83.71 |

*(d)* KD Weight $\lambda_{KD}$

| $\lambda_{KD}$ | Acc. |
|---|---|
| 0.1 | 85.41 |
| 0.5 | 85.33 |
| **1.0** | **88.98** |
| 5.0 | 84.80 |
| 10.0 | 83.59 |

*Table 13.* Sensitivity Analysis of Hyperparameters on TinyImageNet. We evaluate the impact of rectification strength $\gamma$, subspace dimension $k$, and weights for prototype loss ($\lambda_{Proto}$) and distillation loss ($\lambda_{KD}$) on model performance. Default settings are bolded.

*(a)* Rectification Strength $\gamma$

| $\gamma$ | Acc. |
|---|---|
| 0.6 | 50.21 |
| 0.7 | 52.09 |
| **0.8** | **55.54** |
| 0.9 | 50.38 |
| 1.0 | 47.56 |

*(b)* Subspace Dimension $k$

| $k$ | Acc. |
|---|---|
| 1 | 50.44 |
| 3 | 51.27 |
| **5** | **55.54** |
| 8 | 50.04 |
| 10 | 46.28 |

*(c)* Proto Weight $\lambda_{Proto}$

| $\lambda_{Proto}$ | Acc. |
|---|---|
| 0 | 49.52 |
| 0.05 | 51.59 |
| **0.10** | **55.54** |
| 0.50 | 50.02 |
| 1.00 | 49.98 |

*(d)* KD Weigh $\lambda_{KD}$

| $\lambda_{KD}$ | Acc. |
|---|---|
| 0.1 | 49.22 |
| 0.5 | 52.91 |
| **1.0** | **55.54** |
| 5.0 | 53.18 |
| 10.0 | 51.43 |

generative replay, where old classes are synthetic while new classes are real. Once real old-class samples are introduced, this asymmetry is weakened and the shortcut progressively disappears.

### D.6. Additional Hyperparameter Sensitivity

In the main manuscript, we report hyperparameter sensitivity on CIFAR-100. To further validate the robustness of DREAM, we conduct additional ablation studies on CIFAR-10 and TinyImageNet. We vary the rectification strength $\gamma$, subspace dimension $k$, prototype loss weight $\lambda_{Proto}$, and distillation loss weight $\lambda_{KD}$ while keeping the other hyperparameters fixed.

The results in Table 12 and 13 show that the same default configuration, *i.e.*, $\gamma = 0.8$, $k = 5$, $\lambda_{Proto} = 0.1$, and $\lambda_{KD} = 1.0$, works consistently across datasets. Moderate variations around the default setting preserve strong performance, while overly aggressive rectification, such as $\gamma = 1.0$, degrades accuracy. This is consistent with our soft-projection design, since domain-sensitive directions may still contain entangled semantic information and should be suppressed rather than completely removed.

### D.7. Ablation Study on Synthetic Data Quality

Since our method relies on a Training-Free Generator, the quality of synthetic data is a critical factor. We investigate three aspects affecting synthetic quality: the richness of text prompts, the choice of generator architecture, and the usage of data filtering strategies. The results on CIFAR-100 ($T = 10$) are shown in Table 14.

**1. Impact of VLM Prompts:** As discussed in Sec C.1, we extract fine-grained image details—such as category, color, posture, background, and action—via a VLM, which are then used to create text prompts. The format for generating images is: *"A photo of a [CLASS], which has [DETAILS]"*, where DETAILS are extracted by the VLM. Compared to using only category information (*e.g.*, *"A photo of a [CLASS]"*), our method offers richer and more detailed descriptions. The information fed into the generator must be the correct category information. In addition, the final quality of the synthesis image is dependent on the generator. For the same image, we extract fine-grained text using different methods and then feed these texts into the frozen generator. We compare the text results with the image results, as shown in Figure 13. In addition, to further enhance sample quality, we use data filtering to ensure the generative replay data is both accurate and high-quality.

**2. Generator Architecture (Qwen-Image (Wu et al., 2025) vs. SD v1.5 (Rombach et al., 2022)):** We evaluate the impact of changing the underlying generator. While Stable Diffusion v1.5 is a standard baseline (Meng et al., 2024), the more advanced Qwen-Image model generates higher-fidelity images. Interestingly, while Qwen-Image provides a baseline improvement, our rectification method yields significant gains on *both* architectures. This demonstrates that our method is

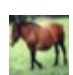 **Qwen2.5-VL** : The image shows the side profile of a brown horse standing outdoors, likely on grass or pasture land. The horse has a sleek coat with a rich reddish-brown coloration, which suggests it might be a breed known for such coloring like an American Quarter Horse or a similar type. Its mane appears short but well-groomed, blending into its bodys hue.The horse's head is turned slightly to one side as if observing something off-camera, giving viewers insight into its alertness. It has dark eyes and ears perked up, indicating attentiveness. The neck curves gracefully from the shoulder down toward the chest area, showcasing the muscular definition typical of horses built for strength and endurance.The background consists mainly of blurred greenery, suggesting a natural setting without any immediate structures visible behind the horse. This gives the impression of being in a rural environment where animals can roam freely. There isnt much detail about the surroundings due to the shallow depth of field used when capturing the photo, focusing attention solely on the horse itself.

**BLIP-2** : This image features a bay horse captured in a side profile, standing and facing to the left against a blurry green grassy background. The horse displays a reddish-brown coat with distinct black points on its mane, tail, and lower legs, and appears to be simply standing in a pasture.

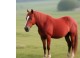

*(a)* CIFAR10

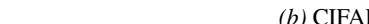

**Qwen2.5-VL** : Bon its shape and posture, there seems to be an animal resembling a cow standing outdoors. The figure has a large body with visible legs, which suggests it could be a bovine It stands upright, possibly grazing as indicated by what looks like grass near its head area. The color scheme includes shades of brown and white, typical for many breeds of cows. Due to the blurriness, specific features such as facial markings, horns (if present), or other distinguishing characteristics cannot be identified accurately.

**BLIP-2** : This image features a cow, likely a Jersey breed with a light brown or tan coat and a dark nose. It is captured in a close-up headshot, wearing a pinkish-red halter and facing directly at the camera. The background is a blurry green pasture with trees in the distance, and the cow appears to be standing calmly while looking toward the viewer.

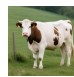 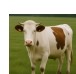

*(b)* CIFAR100

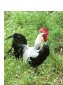 **Qwen2.5-VL** : The image shows a rooster standing on grass, likely in an outdoor setting such as a farm or garden area. The rooster has predominantly black feathers with white markings covering its body, creating a striking contrast against the green background of tall grasses. The roosters head displays vibrant red comb and wattles, which are characteristic of male chickens. It appears to have a strong beak typical for roosters, used for pecking food and defending territory. Its tail feathers are fanned out slightly, showcasing their length and coloration. The legs appear sturdy but not overly large compared to other breeds; they support the bird well while it stands upright. The overall posture suggests alertness, common behavior when a rooster is surveying its surroundings.

**BLIP-2** : This image features a cow, likely a Jersey breed with a light brown or tan coat and a dark nose. It is captured in a close-up headshot, wearing a pinkish-red halter and facing directly at the camera. The background is a blurry green pasture with trees in the distance, and the cow appears to be standing calmly while looking toward the viewer.

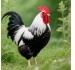 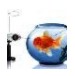

**Qwen2.5-VL** : The image shows a small, vibrant orange fish swimming inside a clear glass bowl with a black base. The fish appears to be a goldfish, which is one of the most popular freshwater aquarium species worldwide due to its hardiness and colorful variety. The goldfish has an oval-shaped body typical for the species. Its scales appear smooth and shiny, reflecting light, giving it a glossy appearance. The fins include two dorsal fins on top, a ventral fin underneath near the tail, and pectoral fins along each side of the body. The tail is slightly rounded or fan-like depending on the specific breed. The bowl itself is simple yet elegant—a transparent glass container with a sleek black stand at the bottom, providing stability while allowing visibility into the water. The simplicity of the setup emphasizes the beauty of the fish without distraction from elaborate decorations. The lighting highlights the golden hue of the fishs scales, making them look vivid against the neutral background.

**BLIP-2** : This image features a goldfish (specifically an Oranda or similar fancy goldfish variety) swimming inside a clear, blue-tinted glass fishbowl. The fish is a vibrant orange-red color and is captured in a side profile, facing to the left. The background is plain white, highlighting the bowl and the fish.

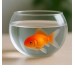 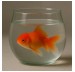

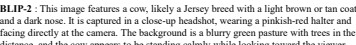

*(c)* ImageNet-Subst

*(d)* TinyImageNet

*Figure 13.* **Robustness to VLM Selection.** We compare the generated results using captions derived from different VLMs (Qwen2.5-VL (Bai et al., 2025) vs. BLIP-2 (Li et al., 2023)). Despite the variation in caption length and detail, the synthesized images remain visually similar and semantically consistent with the original inputs. This indicates that the specific choice of VLM has a negligible impact on the final generation quality.

*Table 14.* Impact of Generator Architecture and Data Filtering. We compare the performance of Qwen-Image versus SD 1.5 and evaluate the efficacy of filtering.

| Generator | Filtering | Data Vol. | Time (h) | Acc (%) |
|---|---|---|---|---|
| SD 1.5 (Rombach et al., 2022) | w/o Filter | 500 | $\approx 5.1$ | 68.08 |
| | w/ Filter | 425 | $\approx 4.1$ | 68.84 |
| Qwen-Image (Wu et al., 2025) | w/o Filter | 500 | $\approx 5.2$ | 68.67 |
| | w/ Filter | 468 | $\approx 4.3$ | **69.73** |

**generator-agnostic**: it effectively rectifies domain bias regardless of the specific artifacts introduced by the generator.

**3. Data Filtering:** We analyze the impact of Task-Constraint Filtering. Since the generated images are of high quality and we focused solely on classification accuracy without considering confidence levels for filtering, only a small number of images were removed. We report the average data volume per category after filtering at each stage.

To assess the sensitivity of our framework to the choice of the Vision-Language Model, we compare the generated samples conditioned on captions from Qwen2.5-VL (Bai et al., 2025) and BLIP-2 (Li et al., 2023). As illustrated in Figure 13, although Qwen2.5-VL provides more granular descriptions compared to the concise outputs of BLIP-2, the resulting synthesized images exhibit high semantic fidelity in both cases. This qualitative comparison suggests that the performance is largely independent of the specific VLM employed, provided the captions capture the core visual attributes.

Consequently, DREAM demonstrates strong robustness to generator architecture. Using the older Stable Diffusion 1.5, our method achieves 68.84%, close to the 69.73% from Qwen-Image, showing that our feature rectification mechanism compensates for varying generative quality. Task-Constraint Filtering consistently improves performance (*e.g.*, boosting Qwen-Image from 68.67% to 69.73%), preventing noisy generations from affecting decision boundaries. However, the filtering effect is modest, indicating that Feature Domain Rectification can "repair" distributions even with low-quality data. Notably, with the same generator (Stable Diffusion 1.5), our method outperforms DiffClass (Meng et al., 2024) by +0.03% (The result of DiffClass is seen in Table 1), even without filtering. This demonstrates that generating data with fine-grained prompts and applying DREAM yields results comparable to methods requiring generator retraining.

Besides, we present examples of the samples generated by the generators for each dataset, as shown in Figure 14.

*Table 15.* **Comparison of Extra Trainable Parameters (Generative Component Only)**. Our method requires no additional gradient-based updates for the generator or auxiliary discriminators. Since DiffClass is not open-source, we can only estimate the number of its additional parameters.

| Method | Generator Adapter | Domain Classifier | Task Classifier | Total Extra Params |
|---|---|---|---|---|
| DiffClass | LoRA ($\approx$ 2M) | Linear Head ($\approx$ 0.00013 M) | ResNet-32 | $\approx$ **2.00013 M** |
| **DREAM (Ours)** | **None (0)** | **None (0)** | ResNet-32 | **0** |

### D.8. Computational Efficiency Analysis

A decisive advantage of our DREAM framework is its **Training-Free generator**. To quantify this efficiency, we conduct a breakdown comparison against the state-of-the-art generative replay method, DiffClass (Meng et al., 2024), focusing on **Trainable Parameter Overhead** and **Training Time Consumption**.

#### D.8.1. TRAINABLE PARAMETER OVERHEAD

In EFCIL, minimizing the memory footprint of additional trainable modules is crucial. We compare the components that require gradient updates:

- **DiffClass:** Requires training a **LoRA** adapter for the diffusion model to learn task-specific distributions, plus a separate **Domain Classifier** to distinguish real/synthetic domains.

- **DREAM (Ours):** Utilizes a completely **frozen** generator. The Feature Domain Rectification (FDR) operates via PCA/SVD on feature matrices, which are non-parametric mathematical operations. No domain classifier is required.

We use ResNet-32 as the classifier for category classification and compare the additional trainable parameters. As shown in Table 15, our method adds **zero** additional parameters for the generative replay component, minimizing storage overhead. Additionally, our trainable parameters are independent of the generator, so even with Stable Diffusion 1.5, our method has fewer trainable parameters than DiffClass (Meng et al., 2024), while achieving comparable or superior performance.

#### D.8.2. TIME CONSUMPTION BREAKDOWN

We further decompose the time cost per incremental task into four components: Generator Training ($T_{train\_gen}$), Image Generation ($T_{gen}$), Domain Classifier Training ($T_{train\_dom}$), and Rectification Calculation ($T_{rect}$, *e.g.*, SVD).

$$T_{total} = \underbrace{T_{train\_gen} + T_{train\_dom}}_{\text{Training Overhead}} + \underbrace{T_{gen}}_{\text{Inference}} + \underbrace{T_{rect}}_{\text{Calculation}} + T_{train\_cls} \tag{42}$$

We estimate the time excluding classifier training time ($T_{train_{cls}}$). Table 16 presents the quantitative comparison.

1. **Training Bottleneck:** DiffClass requires iterative back-propagation to train the LoRA adapter ($T_{train\_gen}$) for every new task, which is the major computational bottleneck. In contrast, our method skips this stage entirely ($T_{train\_gen} = 0$).

2. **Rectification Efficiency:** Since the features used for SVD calculation come from the training cache, no separate extraction is needed. Additionally, the SVD and covariance calculations ($T_{rect}$) are performed once per epoch. Therefore, $T_{rect}$ takes negligible time compared to neural network training.

3. **Inference Trade-off:** While advanced models like Qwen-Image may have a higher single-image inference latency ($T_{gen}$) compared to SD v1.5, the elimination of the training phase results in a significantly lower **Total Time**.

DREAM achieves superior efficiency by eliminating generator training overhead ($T_{train\_gen}$). Unlike DiffClass, which is limited by iterative LoRA fine-tuning, our method incurs negligible cost for rectification ($T_{rect} < 1$ min). Even with the advanced Qwen-Image generator, the total time for DREAM remains comparable to the baseline. Using SD 1.5 saves generator training time entirely, demonstrating an effective balance between performance and training efficiency.

*Table 16.* **Time Consumption Analysis (Per Task on CIFAR-100)**. Time is measured in minutes. Since DiffClass is not open-source, we can only estimate the number of its additional time consumption.

| Method | Backbone | Training Phase | | Generation | Rectification | Total Time |
|---|---|---|---|---|---|---|
| | | $T_{train\_gen}$ (LoRA) | $T_{train\_dom}$ | $T_{gen}$ | $T_{rect}$ (SVD) | |
| DiffClass | SD 1.5 | $\approx 50$ min | $\approx 0$ min | $\approx 50$ min | 0 | $\approx$ **100 min** |
| **DREAM** | SD 1.5 | **0** | **0** | $\approx 50$ min | $< 1$ **min** | $\approx$ **50 min** |
| **DREAM** | Qwen-Image | **0** | **0** | $\approx 100$ min | $< 1$ **min** | $\approx$ **100 min** |

# E. Drawbacks and Future Work

Although DREAM effectively addresses the synthetic-real domain shortcut in training-free generative replay CIL, it still has several limitations that motivate future work.

**Scope of Applicability.** DREAM is designed for standard class-incremental learning under the strict exemplar-free generative replay setting, where old classes are replayed using synthetic images and new classes are observed as real images. This setting induces a clear synthetic-old versus real-new asymmetry, which is the source of the domain shortcut studied in this work. In more symmetric settings, such as joint training or exemplar-based CIL where real and synthetic data are available for both old and new classes, synthetic data may instead serve as beneficial augmentation rather than causing a shortcut. Extending DREAM beyond standard classification CIL remains an important direction.

**Extension to More Complex Scenarios.** Several practical CIL scenarios require additional design beyond the current framework. For long-tailed CIL, generative replay naturally supports class balancing through prompt control, but the interaction between class imbalance and domain rectification needs further study. For detection and segmentation, the current global feature rectification should be extended to spatially structured representations, such as ROI-level or pixel-level alignment. For domain-incremental learning, the domain subspace may evolve over time, requiring dynamic subspace tracking rather than a static projection estimated from the current task.

**Reliance on Shared Domain Subspace.** DREAM relies on the assumption that the frozen generator induces a relatively shared synthetic-real domain subspace across classes. Our theoretical analysis and empirical results support this assumption under the standard single-generator setting, and our additional multi-source experiments show that DREAM remains robust when replay data are generated by heterogeneous sources. However, when the incremental process involves highly diverse generators, drastic real-domain shifts, or continually changing generation styles, the domain bias may become multi-modal and harder to capture using a single static subspace. Future work may explore adaptive rectification, mixture-of-subspaces modeling, or online subspace tracking to better handle such heterogeneous cases.

**Dependence on Foundation Models.** Our generative replay pipeline uses a vision-language model for attribute extraction and a pretrained text-to-image model for image synthesis. As a result, hallucinations from the VLM or biases inherited from the generator may affect replay quality. We mitigate severe semantic errors using task-constraint filtering and reduce isolated feature noise through class-mean centering, but more principled quality control remains valuable. Future work may incorporate uncertainty-aware filtering, confidence calibration, or human-aligned safety constraints to further improve the reliability of generated replay data.

**Computational Trade-offs.** DREAM eliminates generator retraining and introduces no additional trainable generator parameters. The SVD-based rectification itself is lightweight due to feature caching. Nevertheless, the total cost still depends on the inference speed of the frozen generator used to synthesize replay images. More efficient generation strategies, prompt selection, or feature-level replay could further reduce the computational cost while preserving the benefits of training-free generative replay.

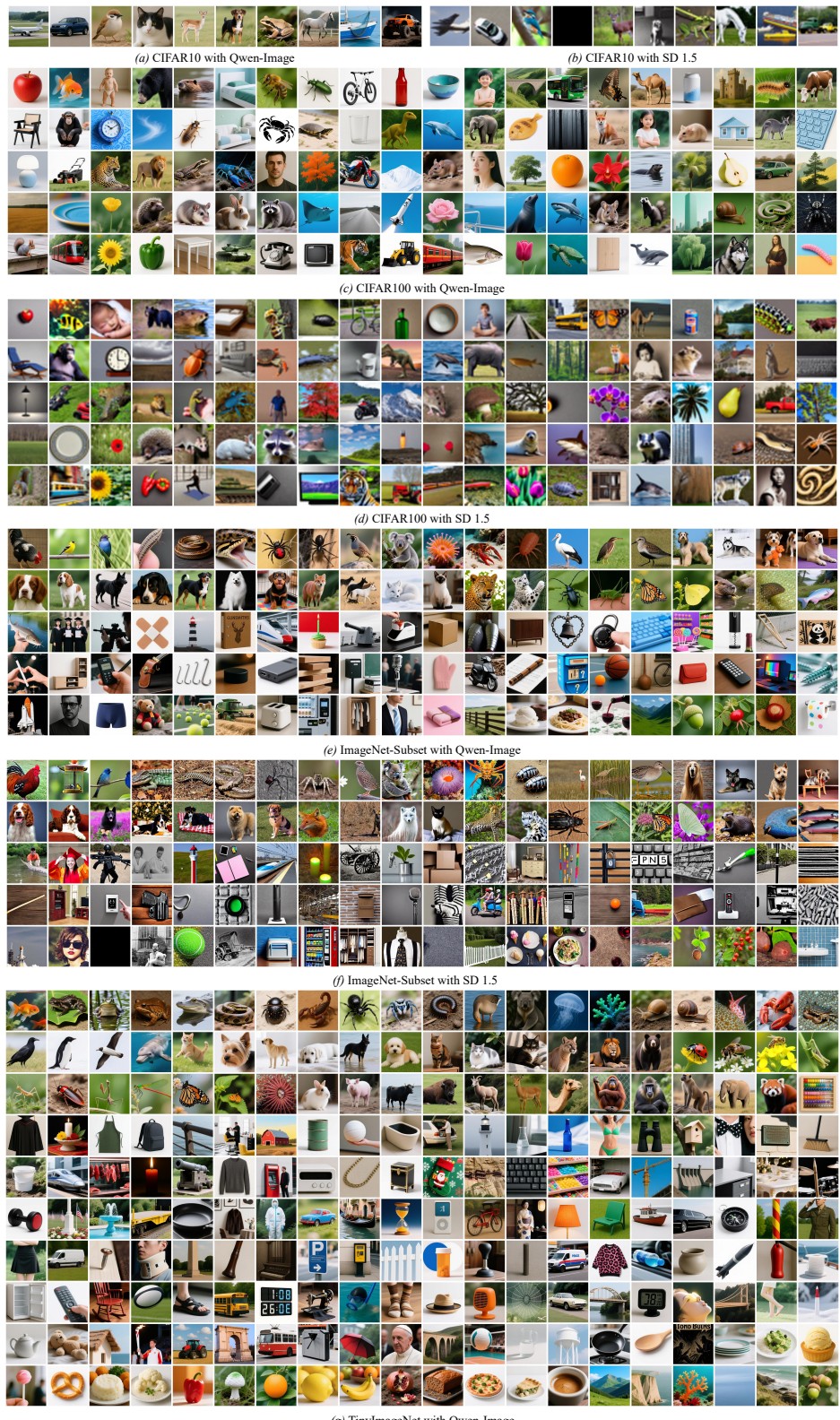

*(a)* CIFAR10 with Qwen-Image      *(b)* CIFAR10 with SD 1.5

*(c)* CIFAR100 with Qwen-Image

*(d)* CIFAR100 with SD 1.5

*(e)* ImageNet-Subset with Qwen-Image

*(f)* ImageNet-Subset with SD 1.5

*(g)* TinyImageNet with Qwen-Image

*Figure 14.* **Visualization of Synthesized Samples.** We display synthetic images generated by our proposed DREAM framework across four standard benchmarks: CIFAR-10, CIFAR-100, ImageNet-Subset, and TinyImageNet. To demonstrate the universality and scalability of our method, we present results using two distinct generative backbones: Qwen-Image and Stable Diffusion v1.5. The visualized samples exhibit high fidelity and diversity, confirming the effectiveness of our approach across different datasets and model architectures.

