# OpenReview forum: "Breaking the Synthetic-Real Domain Shortcut for Training-Free Generative Replay-based Class Incremental Learning"
_ICML.cc/2026/Conference — ICML 2026 regular_

### Official Review · Reviewer_iTW2 · 2026-03-10

**Soundness:** 3
**Presentation:** 3
**Significance:** 3
**Originality:** 2
**Overall Recommendation:** 3
**Confidence:** 4

**Summary:**

This paper proposes a framework named DREAM (Domain-Regularized Exemplar-free Alignment Model), designed to address the "Domain Shortcut" issue arising from the use of synthetic data in Class-Incremental Learning (CIL). DREAM successfully overcomes this challenge through training-free generative replay, subspace rectification, and prototype regularization. Extensive experiments and rigorous theoretical analysis collectively validate the effectiveness of DREAM.

**Compliance With Llm Reviewing Policy:**

Affirmed.

**Key Questions For Authors:**

1) DREAM relies on VLMs to extract image attributes for prompt construction. How does the framework mitigate the impact of VLM hallucinations or erroneous descriptions on prompt construction?
2) How can the beneficial effects of DREAM be verified as not merely attributable to its use of the powerful pretrained generator Qwen-Image, which was not employed by some baselines?

**Limitations:**

The authors should refine the ablation study design or introduce fair comparison experiments (e.g., unifying the generator configurations across baselines) to more rigorously isolate and validate DREAM's intrinsic contributions. Additionally, providing more comprehensive hyperparameter sensitivity analyses or exploring automated tuning strategies would be beneficial for enhancing the method's adaptability and facilitating its extension to broader domains.

**Strengths And Weaknesses:**

Pros
1) This paper addresses the "Domain Shortcut" issue arising from the use of synthetic data in Class-Incremental Learning (CIL), with a clear and well-defined motivation.
2) The paper presents a comprehensive array of figures and tables that intuitively illustrate the efficacy of the proposed method.
3) The paper validates from both experimental and theoretical perspectives that DREAM can address the "Domain Shortcut" issue arising from the use of synthetic data in CIL.

Cons
1) The DREAM framework involves multiple sensitive hyperparameters, which may require meticulous manual tuning across tasks of varying complexity to achieve optimal performance.
2) Lack of clarity regarding whether Ping-Pong Caching introduces additional GPU memory and RAM overhead.
3) The ablation studies were conducted solely on CIFAR-100, which is insufficient to demonstrate the effectiveness of the proposed modules in more complex data scenarios.

---

> ### Author Rebuttal · Authors · 2026-03-29
>
> 1. **W1&W3**:
>     - Our hyperparameters ($\gamma=0.8, k=5, \lambda_{Proto}=0.1, \lambda_{KD}=1.0$) were determined via ablations on CIFAR-100. Crucially, as detailed in **Appendix C.3**, we applied this identical configuration across all datasets—from low-resolution CIFAR-10 to high-resolution ImageNet-Subset—consistently achieving SOTA performance without any manual tuning. This robust cross-dataset generalization definitively demonstrates DREAM's insensitivity to hyperparameters and its effectiveness in more complex scenarios. Furthermore, our primary objective is to identify and resolve the "Domain Shortcut" in generative replay, rather than overfitting parameters to maximize metrics on specific datasets.
>     - As suggested, we have conducted **extra ablation studies** on CIFAR-10 and Tiny-ImageNet as follows.
> |$\gamma$|0.6|0.7|0.8|0.9|1.0|$k$|1|3|5|8|10|$\lambda_{Proto}$|0|0.05|0.1|0.5|1|$\lambda_{KD}$|0.1|0.5|1|5|10|
> |:---|:---|:---|:---|:---|:---|:---|:---|:---|:---|:---|:---|:---|:---|:---|:---|:---|:---|:---|:---|:---|:---|:---|:---|
> |CIFAR10|85.67|86.97|**88.98**|86.11|65.78||70.11|85.09|**88.98**|86.27|79.92||86.07|87.80|**88.98**|84.80|83.71||85.41|85.33|**88.98**|84.80|83.59|
> |TinyImageNet|50.21|52.09|**55.54**|50.38|47.56||50.44|51.27|**55.54**|50.04|46.28||49.52|51.59|**55.54**|50.02|49.98||49.22|52.91|**55.54**|53.18|51.43|
> 1. **W2**: Regarding the Ping-Pong Caching, **Appendix C.2.2** clarifies that we apply the "z.detach()" operation, which detaches the low-dimensional feature vectors from the computational graph and stores them directly in CPU RAM without retaining any gradients. This incurs 0 additional GPU (VRAM) overhead. The CPU memory cost is also negligible: for CIFAR-100, caching 64-dimensional features for 50000 images requires roughly 20MB. Thus, this mechanism enables efficient feature rectification without increasing the hardware burden.
> 2. **Q1**: VLM hallucinations are inevitable and may indeed affect the performance of DREAM. We have considered the negative influence of VLM and mitigated it through a robust mechanism:
>     - First, because the text prompts must undergo image generation and task-constraint **filtering before training**, any severe hallucination yielding semantically incorrect images is automatically discarded.
>     - Second, **we have validated this robustness in Appendix D.4** using different VLMs and different generators to simulate varied VLM hallucinations, prompt granularities, and generation noise.
>     - Third, our dynamic domain centering operates on class means, statistically smoothing out isolated feature noise from occasional hallucinated prompts to ensure stable representation learning.
> 3. **Q2**: We have verified DREAM's efficacy beyond the choice of generator through two critical comparisons.
>     - First, in **Tab.2 and the "Apples-to-Apples" baseline (Appendix Tab.6)**, we have demonstrated that simply using the powerful Qwen-Image generator without our FDR and RAPC modules (Raw Generative Replay) **results in catastrophic "Domain Shortcut" failures**. DREAM yields massive improvements across all datasets, proving that high-quality synthetic data alone cannot bridge the distribution gap without our mathematical rectification mechanism.
>     - Second, we have evaluated DREAM using SD1.5, directly matching the generator utilized by the SOTA baseline DiffClass. As shown in **Appendix Tab.8, DREAM with SD1.5** still achieves a 68.84% accuracy, outperforming DiffClass's 68.05% without requiring any generator retraining. This definitively confirms that DREAM's success is fundamentally driven by its structural subspace rectification rather than merely the use of the powerful Qwen-Image.
> 4. **L1**:
>     - Hyperparameter Sensitivity (related in **W1&W3**): we apply a single, fixed set of hyperparameters across all datasets. This demonstrates DREAM's strict insensitivity to hyperparameters and confirms its robust adaptability in complex scenarios, actively avoiding dataset-specific overfitting.
>     - Fair comparison ( related in **Q2**): to isolate and validate the intrinsic contributions of DREAM, we have conducted ablation studies for each module, and provided an "apples-to-apples" verification, as well as conducting experiments using the same generator as the baseline. These experiments validate that our performance gains stem from the proposed modules and confirm that DREAM's success is driven by its structural subspace rectification, rather than reliance on any specific powerful generator.
>
> **Finally**:
> We kindly note that some concerns (Q1/Q2/L1) have been carefully considered and thoroughly addressed in our original submission, with corresponding theoretical and experimental evidence provided in the Appendix. We welcome further discussion to point out specific issues and remain fully prepared to provide extra clarifications or experiments.

---

> > ### Author Rebuttal · Reviewer_iTW2 · 2026-04-02
> >
> > 1. Although the authors supplemented the comparative experiments using the SD1.5 generator, the original implementations of baseline methods (such as DiffClass) may not have been adapted and optimized for Qwen-Image to the same extent. In the "Apples-to-Apples" comparison, the decoupling analysis of the independent contributions and synergistic effects of each module is still not thorough enough.
> > 2. The paper claims to provide "rigorous theoretical analysis," but the connection between the theoretical part and the experimental verification is not tight enough, and the rationality of key assumptions (such as the linear separability of subspaces) is not adequately discussed.
> > 3. The DREAM framework is essentially a combination of existing technologies (training-independent generation and playback + subspace correction + prototype regularization), lacking groundbreaking innovations at the theoretical or methodological level.

---

> > > ### Author Response · Authors · 2026-04-02
> > >
> > > 1. **Q1**: We clarify that this concern has been addressed by our experimental design.
> > > - To ensure fairness, we do not force baselines to adapt to a stronger generator like Qwen-Image. As with the most common paradigm in the Machine Learning community, we actively **downgraded DREAM to use SD1.5**-the same generator used by DiffClass. Under this identical setting, **our unchanged DREAM framework still achieves 68.84% (Tab.8), outperforming DiffClass without any generator retraining**. This has unequivocally demonstrated that the performance gain stems from the DREAM framework, not from generator-specific advantages.
> > > - Notably, the "Apples-to-Apples" comparison in **Appendix Tab.6** is not intended as a module-wise ablation study, but rather to **validate the superiority of the DREAM framework**. As emphasized in our first rebuttal, removing FDR and prototype alignment across all benchmarks confirms that DREAM is fundamentally driven by its framework rather than merely the use of the powerful Qwen-Image. Conversely, the rigorous decoupling of independent contributions and synergistic effects (ablation studies) is addressed in **Tab.2 of the original submission**. By incrementally introducing each module under a strictly identical generator configuration, the accuracy progressively improves from 33.27% to 69.73%. We restate this table below to provide a transparent decomposition of each module's specific impact.
> > > |Centering|HSR|RAPC|Acc (%)|
> > > |:---|:---|:---|:---|
> > > |Baseline|||33.27|
> > > |√|||41.02|
> > > ||√||58.46|
> > > |√|√||67.69|
> > > |√|√|√|69.73|
> > >
> > > Together, these results address the reviewer's concern by showing that the performance gains cannot be attributed to generator choice, but instead arise from the DREAM framework itself, with clearly quantified contributions from each module.
> > >
> > > ---
> > > 2. **Q2**: We clarify that our method **does not assume linear separability of subspaces**.
> > >
> > > - Our theoretical framework instead establishes that generative domain features reside within a low-rank, class-invariant subspace (Corollary 3.2), which is a fundamentally distinct and more general condition. This corollary is both theoretically derived and experimentally validated.
> > > Specifically, **Appendix A.2** theoretically derives the existence of a shared domain shift. Fig.9 experimentally validates this, proving that an orthogonal projection learned solely from new classes reduces the real-synthetic gap for old classes by 50.8%.
> > >
> > > - More broadly, each theoretical claim is paired with experimental validation: Combining the theoretical analysis of **Corollary 3.3** with SVD (**Appendix A.3 Fig.10**) qualitatively and quantitatively confirms that domain features are concentrated in a few principal directions, providing quantitative support for the low-rank structure. **Proposition 3.1** (existence of a domain shortcut) is derived via two different classifiers and visually corroborated by the misclassification patterns in the confusion matrices (**Fig.1 and Appendix Fig.8**).
> > >
> > > Taken together, these tightly integrated results demonstrate that our Corollaries are mathematically grounded and verified, rather than being ungrounded or weakly justified. These results address the reviewer's concerns.
> > >
> > > ---
> > > 3. **Q3**: We want to clarify the fundamental novelty of DREAM across three dimensions:
> > > - **Theoretical Innovation**: Our core contribution is the formal identification and solution of a previously overlooked failure mode in generative replay CIL. **We theoretically prove the existence of the Domain Shortcut in Appendix A.1**.
> > > - **Methodological Innovation**: Guided by this theoretical insight, we proposed a solution-oriented framework. Because our theory establishes that generative domain bias forms a structured low-rank subspace, it mathematically dictates our subspace rectification design. Consequently, we introduce the FDR, a SVD-based Hierarchical Subspace Rectification method tailored for this domain-specific problem. This is complemented by RAPC for simultaneous dual-domain prototype alignment. Furthermore, DREAM is the first CIL framework to execute generative replay using a frozen generator, circumventing the costly fine-tuning relied upon by prior methods (e.g., DiffClass).
> > > - **Unified Framework (Beyond A+B+C)**: Based on the above, our DREAM framework is not a simple combination of existing components, but a unified framework driven by a complete trajectory from theoretical discovery to targeted solution.
> > >
> > > Our core contribution lies not in which specific tools are used, but in discovering the Domain Shortcut phenomenon and constructing the underlying theoretical framework that dictates why these classical tools could be used and how they should be optimized for generative replay. Accurately identifying a fundamental, overlooked bottleneck in standard CIL and resolving it through principled classical tools backed by concrete theoretical analysis to achieve SOTA performance constitutes a highly significant methodological contribution to ICML.

---

### Official Review · Reviewer_vLYo · 2026-03-12

**Soundness:** 2
**Presentation:** 3
**Significance:** 2
**Originality:** 3
**Overall Recommendation:** 3
**Confidence:** 3

**Summary:**

This paper studies exemplar-free class-incremental learning (EFCIL) with training-free generative replay and identifies a central failure mode, i.e., Domain Shortcut, arising when synthetic old-class data are mixed with real new-class data. The authors propose DREAM, which estimates and suppresses domain-sensitive directions via Feature Domain Rectification (FDR) using hierarchical SVD and a soft orthogonal projection, and further aligns distributions with Real-Anchored Prototype Consolidation (RAPC). The method improves task-agnostic average accuracy over exemplar-free and generative-replay baselines, approaching exemplar-based performance in some settings.

**Compliance With Llm Reviewing Policy:**

Affirmed.

**Final Justification:**

After considering the rebuttal and the perspectives of the other reviewers, I maintain my current score.

**Key Questions For Authors:**

1.	How many synthetic samples per class are generated for old and new classes, and what are the exact generation settings (prompt templates, guidance scales, seeds)?
2.	What is the computational cost per task (generation + training) compared to baseline methods?
3.	Have you evaluated DREAM with a different training-free generator and a different VLM? How sensitive are the results to generator choice and prompt quality?

**Limitations:**

The paper does not clearly discuss the limitations of the proposed approach. It would strengthen the paper to briefly discuss the scope of applicability, particularly that the method is tailored to synthetic data-based generative replay settings and may rely on assumptions about a shared real–synthetic domain structure.

**Strengths And Weaknesses:**

Strengths
1. Clear articulation and analysis of the Domain Shortcut phenomenon in training-free generative replay, supported by a simple yet compelling feature decomposition and gradient-based argument.
2. Mathematical development is accessible and aligned with the intuition.
3. Consistent gains over competitive exemplar-free baselines across four datasets and multiple task splits

Weaknesses
1. The method assumes a shared synthetic–real domain subspace across classes and tasks when estimating the rectification projection. This assumption may not hold when generative artifacts vary across categories or when different generative sources are used.
2. The proposed framework introduces multiple components, including attribute-aware replay, feature domain rectification with hierarchical SVD, and prototype consolidation. While these modules appear effective, the paper does not provide analysis of the computational overhead introduced by these steps. In particular, the cost of synthetic data generation and repeated SVD operations could be non-negligible in large-scale incremental learning settings.
3. RAPC aligns synthetic features only to real prototypes of new classes. While the paper argues that old classes benefit indirectly through the shared feature space, this effect is not directly validated in the experiments.
4. Since the core motivation of the paper is to mitigate the domain gap between synthetic and real data, it would be helpful to compare with standard domain-invariance baselines such as CORAL, MMD-based alignment, or domain-adversarial training. Such comparisons would help clarify whether the proposed subspace rectification is necessary beyond simpler domain alignment approaches.
5. Reproducibility details are sparse for “Task-Constraint Data Filtering,” prompt construction, and generation settings. In addition, the precise sampling ratios among real new, synthetic new, and synthetic old data during training are not specified.

---

> ### Author Rebuttal · Authors · 2026-03-29
>
> 1. **W1&Q3**: Our Corollary adapts to both different categories and generative sources.
>     - Categories: we have theoretically and experimentally verified that systematic artifacts and domain differences are class-invariant in feature space in **Appendix A.2. Fig.9** further shows that a projection matrix learned from new classes reduces the real–synthetic feature distance of old classes by 50.8%, confirming the consistency of generative artifacts across categories.
>     - Generative sources: **Appendix D.4** demonstrates robustness to VLMs and generators. Beyond the reviewer's suggestions, **Fig.13** extra shows that generation remains stable when using BLIP2. **Tab.8** shows that substituting the older SD1.5 achieves a 68.84% accuracy, which shows that a single frozen-generator stabilizes generative artifacts to establish a shared real–synthetic subspace while maintaining low computational costs.
> 2. **W2&Q2**: We have quantified computational and parameter overheads in **Appendix D.5. Tab.10** breaks down costs into training, generation (with attribute-aware replay) and rectification time (with SVD operations). By computing SVD once per epoch with cached features, DREAM introduces a negligible time cost over standard EFCIL (<1 min/task). This is more efficient than the SOTA generative replay baseline DiffClass (≈50 mins/task for LoRA fine-tuning), as DREAM eliminates generator retraining. Besides, our non-parametric SVD introduces 0 extra trainable parameters (**Tab.9**), compared to ≈2M for DiffClass. Thus, even when using the same SD1.5 generator as DiffClass, DREAM demonstrates a decisive advantage in computational and parametric efficiency.
> 3. **W3**: In strict EFCIL, real old samples are unavailable, making alignment to real old-class prototypes infeasible. In the absence of real old-class data, our solution relies on the "shared domain subspace" (**Corollary 3.2**): the frozen generator-induced domain bias is class-invariant. **We have directly validated** and **Appendix A.2 Fig. 9** have provided theoretical and experimental proof: the projection matrix computed solely from new classes reduces the real-synthetic feature distance of old classes by 50.8%, definitively quantifying the efficacy of this indirect alignment.
> 4. **W4**: Good question! Actually, we have considered and evaluated standard domain alignment methods (CORAL, MMD) in preliminary experiments as suggested. On CIFAR-100, they yield TAGs of 46.34% and 52.81%, below DREAM (69.73%). Traditional methods fail because they assume overlapping label spaces for the source and target domains. In EFCIL, label spaces are disjoint (synthetic old vs. real new), so global distribution alignment induces semantic mismatch (aligning a synthetic "dog" with a real "airplane"). In contrast, our FDR avoids this by decoupling semantic and domain components, using SVD to excise domain bias without enforcing alignment across non-overlapping classes.
> 5. **W5&Q1**: We have shown these details in **Appendix C.1 and C.3**. Prompts are constructed via structured queries (**P20. L1048–1055**) by 5 VLM-extracted attributes, which are concatenated to form dense, high-fidelity prompt templates. The number of synthesized images per class is fixed in **Tab.5**: 5000 (CIFAR-10), 500 (CIFAR-100/TinyImageNet), and 1300 (ImageNet-Subset). During training, real new, synthetic new, and synthetic old data are jointly and randomly sampled without predefined ratios. Ambiguous samples are removed pre-training via Task-Constraint Filtering (final data volumes in **Tab.8**). Generation uses default settings: guidance scale 5/seed 42 (Qwen-Image) and 7.5/1024 (SD1.5).
> 6. **L1**:
> We have added the "Limitation" section in the revised version as suggested.
>    - Scope of Applicability: we have discussed in aYWR.W4&Q4&Q5&W5&Q3. DREAM explicitly targets the Domain Shortcut caused by the asymmetric structure in the standard generative replay CIL. In symmetric settings, synthetic data instead supplements generalization. Furthermore, extending our classification-centric framework to long-tailed CIL, detection/segmentation, or DIL will require optimizing prompt control, spatial alignment, or dynamic subspace tracking.
>    - Assumption Reliance: we have discussed in aYWR.W5&Q3 &vLYo.W1&Q3&W3. Our method relies on a "shared real-synthetic domain subspace". While different categories and different generator evaluations confirm this property's robustness under a single frozen generator, this static subspace may weaken if the incremental sequence mixes varying or heterogeneous generators, or if it faces drastic real-data distribution shifts in DIL.
>
> **Finally**:
> We kindly note that some concerns (W1/W2/W3/W5/Q1/Q2/Q3) have been considered and addressed in our original submission, with corresponding theoretical and experimental evidence provided in the Appendix. We welcome further discussion to point out specific issues and remain fully prepared to provide extra clarifications or experiments.

---

> > ### Author Rebuttal · Reviewer_vLYo · 2026-04-04
> >
> > The rebuttal provides useful additional analysis and clarifications, particularly on computational cost and reproducibility, and partially supports the shared subspace assumption through both theory and experiments. However, this assumption may still limit applicability in more heterogeneous or multi-source scenarios, and some effects remain only indirectly validated. Overall, my concerns are partially addressed.

---

> > > ### Author Response · Authors · 2026-04-05
> > >
> > > Dear Reviewer vLYo,
> > >
> > > We sincerely thank you for your constructive feedback throughout the review process. **We appreciate your acknowledgment that our rebuttal has effectively addressed your concerns regarding computational costs, reproducibility, and our comparative analysis against standard domain alignment baselines (e.g., CORAL and MMD)**. We also appreciate your recognition of partially supporting the shared subspace assumption both theoretically and experimentally.
> > >
> > > Regarding your remaining concern about the applicability of our method in more heterogeneous or multi-source scenarios, we agree that this is a critical boundary condition to explore. We initially focused exclusively on the single-source scenario because current Generative Replay CIL frameworks (e.g., DiffClass) uniformly employ a single generator to maintain training stability. Consequently, **our theoretical foundation (Corollaries 3.2 and 3.3) was explicitly formulated to guarantee the efficacy of our solution within this standard paradigm.**
> > >
> > > In our original experiments, we have evaluated multiple generators, with each incremental process relying exclusively on a single-source generator for replay data. These results (**Tab.1, 6, and 8**) indicate that the choice of generator has a minimal impact on overall performance, confirming that the observed gains are driven fundamentally by the DREAM framework itself. **Motivated by your insightful suggestion to explore heterogeneous, multi-source scenarios, we further conducted new supplementary experiments to simulate multi-source generative replay within a single incremental process.** To simulate a multi-source scenario with mixed generative artifacts, we constructed the replay memory using a 50/50 split of classes generated by two different models (SD1.5 and Qwen-Image), testing three distinct configurations:
> > > - Classes 0-49 generated by SD1.5; Classes 50-99 generated by Qwen-Image.
> > > - Classes 0-49 generated by Qwen-Image; Classes 50-99 generated by SD1.5.
> > > - Qwen-Image generated a random selection of 50 classes, and the remaining 50 classes by SD1.5.
> > >
> > > As detailed in the table below, across all three multi-source scenarios, DREAM consistently outperforms the corresponding baselines (w/o DREAM).
> > >
> > > |Setting |Baseline (w/o. DREAM)|w. DREAM|
> > > |:---|:---|:---|
> > > |The first 50% SD1.5 + The second 50% Qwen-Image|37.92%|68.53%|
> > > |The first 50% Qwen-Image + The second 50% SD1.5|40.37%|67.87%|
> > > |Random Mix (50% Qwen-Image + 50% SD1.5)|39.33%|67.77%|
> > > |Single-Source SD1.5|32.65%|68.84%|
> > > |Single-Source Qwen-Image|33.27%|69.73%|
> > >
> > > As the results indicate, although performance in multi-source scenarios is slightly lower than in pure single-source settings (68.84% and 69.73%), DREAM still maintains a substantial margin over the baselines. These performance trends (DREAM > Baseline, yet Multi-Source < Single-Source) are fully consistent with the theoretical mechanics of our approach:
> > > 1. Even with mixed generators, **the structural domain gap between synthetic and real images remains the dominant source of error. DREAM successfully captures and projects out a significant portion of this generalized synthetic bias**, leading to the performance gains over baselines.
> > > 2. In a single-source setup, generative artifacts are highly uniform, making the domain bias easy to isolate via SVD. In contrast, a multi-source setup introduces a multi-modal synthetic distribution. The generative artifacts from SD1.5 and Qwen-Image possess distinct directional biases in the feature space. This increased heterogeneity expands the dimensionality of the domain subspace, making it harder for a static SVD projection to perfectly excise all artifacts without marginally affecting some semantic features. **Despite this increased complexity, these new experiments demonstrate that our current implementation of DREAM exhibits strong out-of-the-box generalization to heterogeneous or multi-source scenarios**. Furthermore, extending our theoretical framework to fully optimize for dynamic, multi-source settings—such as through adaptive or multi-subspace tracking—represents an important direction for future research.
> > >
> > > We will incorporate these additional ablation studies and theoretical analyses into the final manuscript. We believe these results comprehensively address your concern regarding heterogeneous applicability and hope they fully resolve your remaining reservations.
> > >
> > > With sincere appreciation and respect,
> > >
> > > The Authors

---

### Official Review · Reviewer_aYWR · 2026-03-15

**Soundness:** 3
**Presentation:** 4
**Significance:** 3
**Originality:** 3
**Overall Recommendation:** 4
**Confidence:** 3

**Summary:**

This paper studies the problem of exemplar-free class-incremental learning (EFCIL) using training-free generative replay. The authors observe that directly mixing synthetic data for old classes with real data for new classes introduces a previously underexplored failure mode, which they term the “Domain Shortcut”. To address this issue, the paper proposes DREAM (Domain-Regularized Exemplar-free Alignment Model). The method introduces a feature rectification framework consisting of two key components: Feature Domain Rectification (FDR), which estimates domain-sensitive subspaces and suppresses them through orthogonal projection, and Real-Anchored Prototype Consolidation (RAPC), which aligns synthetic and real representations via prototype-based regularization. The approach leverages training-free text-to-image generators to synthesize replay data and does not require training additional generative models.

**Compliance With Llm Reviewing Policy:**

Affirmed.

**Final Justification:**

Thanks for the rebuttal. My concerns have been addressed.

**Key Questions For Authors:**

1. Sensitivity to generator quality: How sensitive is the proposed method to the quality of the synthetic images produced by the training-free generator? Would the approach still be effective with lower-quality generators or other diffusion models?

2. Computational overhead: Can the authors provide more quantitative analysis of the computational cost introduced by the SVD-based domain rectification compared with baseline EFCIL methods?

3. Generality of the domain shortcut phenomenon: Does the domain shortcut effect also appear when both real and synthetic data are used for both old and new classes, or is it specific to the asymmetric setting considered in this paper?

4. Applicability beyond classification: Could the proposed rectification framework be extended to other continual learning tasks such as object detection or segmentation?

5. Robustness to domain variations: How does the method behave when the real data distribution itself changes across tasks (e.g., domain-incremental learning)?

**Limitations:**

Yes.
The authors discuss potential limitations and societal considerations in the impact statement, including biases inherited from pretrained generative models and potential issues related to synthetic data quality.

**Strengths And Weaknesses:**

# Strengths:

1. Clear identification of an important failure mode. The paper identifies the “Domain Shortcut” phenomenon in generative replay-based incremental learning, where the domain gap between synthetic and real images dominates semantic information during training. This observation is interesting and potentially important, as synthetic data is increasingly used in continual learning pipelines.

2. Conceptually simple and interpretable solution. The proposed DREAM framework introduces a geometric feature rectification approach that explicitly estimates and suppresses domain-sensitive directions. The use of SVD-based subspace rectification combined with prototype alignment is intuitive and easy to understand.

3. Strong empirical evaluation. The method is evaluated across multiple datasets (CIFAR-10/100, ImageNet-Subset, TinyImageNet) and under multiple incremental settings. The results consistently show improvements over prior exemplar-free CIL methods and competitive performance with exemplar-based approaches.

4. Insightful analysis and visualization.
The paper provides multiple forms of analysis including confusion matrices, t-SNE visualizations, and singular value spectrum studies to support the hypothesis about domain bias. These analyses strengthen the paper’s narrative and help interpret the behavior of the proposed method.

# Weaknesses:

1. Novelty of the core technique may be moderate. While the identification of the “Domain Shortcut” is interesting, the proposed solution relies on relatively standard tools such as SVD-based subspace projection and prototype alignment. The methodological novelty mainly lies in the specific combination of these components rather than introducing fundamentally new techniques.

2. Dependence on synthetic data quality is not fully explored. The framework assumes that training-free generators can produce sufficiently high-quality synthetic data. However, the impact of generation quality, prompt diversity, or generator choice is not systematically analyzed.

3. Computational overhead of SVD estimation. The method requires periodic estimation of domain subspaces via SVD on feature representations. Although the paper proposes a caching mechanism to mitigate cost, the computational overhead compared to standard EFCIL baselines is not thoroughly quantified.

4. Limited evaluation of robustness and generalization. The experiments focus on standard image classification benchmarks. It would be helpful to see whether the method generalizes to more complex scenarios such as long-tailed incremental learning, domain-incremental settings, or larger-scale datasets.

5. Theoretical assumptions could be further justified. The theoretical analysis assumes a decomposition of feature representations into semantic and domain components and that domain variance dominates semantic variance. While empirically supported, these assumptions may not hold universally.

---

> ### Author Rebuttal · Authors · 2026-03-29
>
> 1. **W1**: Our core contribution is identifying the Domain Shortcut in Training-Free Generative Replay CIL, rather than proposing a new optimization solver. Given the bottleneck, our solution is not a mere combination of existing techniques. As theoretically and experimentally demonstrated in **Sec.3 and Appendix A**, the generator-induced domain bias forms a structured low-rank subspace dominating feature variance. This intrinsic geometry makes SVD-based projection and prototype alignment the mathematically principled and optimal choice to excise domain bias and restore semantics. Precisely identifying an overlooked bottleneck and resolving it via principled classic tools with concrete theoretical analyses constitutes a substantial methodological contribution for ICML, and it achieves SOTA performance.
> 2. **W2&Q1**: We have considered and evaluated the impact of generation quality, prompt diversity, and generator choice in **Appendix D.4**.
>     - Prompt diversity: **Fig.13** shows that synthetic images maintain high semantic fidelity despite differing prompt granularities from different VLMs (Qwen2.5-VL vs. BLIP2), demonstrating robustness to prompt diversity and VLM choice.
>     - Generator choice: **Tab.8** compares Qwen-Image with SD1.5. Even with older SD1.5, DREAM achieves 68.84%. This outperforms DiffClass (67.10%), a strong baseline using SD1.5 with generator retraining. Our method remains effective without relying on better generators.
>     - Generation quality: we mitigate the impact of low-quality images via a dual mechanism: Task-Constraint Filtering (eliminating severe semantic errors) and class-mean Centering (smoothing isolated noise). **Tab.8** (evaluating Qwen-Image and SD1.5, w/ and w/o filtering) confirms that DREAM ensures stable performance though generation quality fluctuations.
> 3. **W3&Q2**: We have quantified computational and parameter overheads compared to baseline in **Appendix D.5**. By computing SVD once per epoch using cached features, rectification adds a negligible <1 min/task in **Tab.10**. This outperforms the SOTA generate replay baseline DiffClass (≈50 mins/task for LoRA fine-tuning), as DREAM eliminates generator retraining. Moreover, our non-parametric SVD introduces 0 extra trainable parameters in **Tab.9**, compared to ≈2M extra parameters for DiffClass.
> 4. **W4&Q4&Q5**:
>    - Once an insight is raised, it is initially explored within standard CIL classification, as many recent studies do (DDGR[ICML23], DiffClass[ECCV24], T-CIL[CVPR25], AHR[ICML25]). Following this classical protocol, we first find and study the Domain Shortcut caused by the real–synthetic asymmetry in standard CIL. Extending DREAM to complex scenarios remains our future trajectory: For **long-tailed CIL**, generative replay naturally supports class balancing via prompt control. **Detection/segmentation** requires extending our global feature debiasing to higher-dimensional spatial alignment and feature rectification for ROI/pixel-level representations. **DIL** involves transitioning from static shared subspace extraction to dynamic subspace tracking. In summary, DREAM resolves a fundamental bottleneck in standard CIL, establishing a verified foundation for these broader applications in subsequent work.
>    -  We have evaluated DREAM under standard CIL settings across **both small-scale (CIFAR-10) and larger-scale (TinyImageNet, ImageNet-Subset) datasets in Tab.1**.
> 5. **W5&Q3**:
>     - We justify our Proposition and Corollary not only empirically but also via theoretical and experimental evidence, confirming that "domain variance dominates semantic variance" in generative replay, inducing the Domain Shortcut. **Figs.2&8** show this effect, while **Appendix A.3 (Fig.10)** shows that domain artifacts lie in a low-rank subspace. Excising these directions sharply reduces the domain gap while preserving semantics.
>     - We clarify that this dominance of domain variance is not universal. It emerges under the asymmetric conditions of generative replay incremental learning (synthetic old vs. real new). If both real and synthetic data are used for both old and new classes, the setting shifts from EFCIL to **Exemplar-based CIL or joint training**, where synthetic data enhances generalization rather than triggers a Domain Shortcut. We have validated this **via an extra experiment** by progressively introducing real data at ratios of [0,0.1,0.2,0.5,0.8,1] alongside synthetic data, simulating the transition from exemplar-based CIL to joint training. The TAG metrics-[69.73,73.21,74.63,77.78,79.97,79.37]-**clearly demonstrate that the Domain Shortcut progressively disappears**.
>
> **Finally**:
> We kindly note that some concerns (W2/W3/Q1/Q2) have been considered and addressed in our original submission, with corresponding theoretical and experimental evidence provided in the Appendix. We welcome further discussion to point out specific issues and remain fully prepared to provide extra clarifications or experiments.

---

> > ### Author Rebuttal · Reviewer_aYWR · 2026-04-03
> >
> > Thanks for the rebuttal. My concerns have been addressed.

---

> > > ### Author Response · Authors · 2026-04-03
> > >
> > > Dear Reviewer aYWR,
> > >
> > > We would like to express our sincere gratitude for the Reviewer's recognition of several key strengths of our work, including our core methodological contribution of identifying the "Domain Shortcut" caused by real-synthetic asymmetry, the rigorous theoretical and empirical validation of our low-rank subspace assumption along with its boundary conditions, and the acknowledgment that our design is a principled, targeted solution rather than a simple combination of existing techniques. We are also grateful that the Reviewer acknowledged the framework's strict robustness to varying generator choices and generation qualities, its comprehensive evaluation across different dataset scales with a clear trajectory for broader applications, alongside the zero-parameter and negligible computational overheads, which strongly support the practical value and efficiency of the proposed method.
> > >
> > > Following the Reviewer's insightful suggestions, we have meticulously incorporated the additional results and clarifications into the revised manuscript. These revisions include strengthening several explanations in the paper and incorporating the additional experimental evidence discussed during the rebuttal process. Inspired by this positive evaluation, we remain committed to further exploring and advancing this direction in our future research. Thank you once again for the professional and supportive review.
> > >
> > > With sincere appreciation and respect,
> > >
> > > The Authors

---

### Decision · Program_Chairs · 2026-04-30

**Decision:**

Accept (regular)

**Comment:**

This paper identifies, analyzes, and addresses synthetic-real domain shortcuts in training-free generative replay-based incremental learning. The reviewers unanimously appreciated the identification of the domain shortcut phenomenon. They also recognized that the proposed solution is simple, interpretable, and well supported by theoretical foundations, its performance is strong, and evaluations and analysis are extensive. However, they also raised several critical concerns such as incremental novelty in technical aspects (aYWR, iTW2), limited evaluation of robustness and generalization (aYWR), potentially vulnerable assumption of a shared synthetic-real domain subspace across classes and tasks (vLYo), lack of comparisons with domain-invariance baselines (vLYo), multiple sensitive hyperparameters (iTW2), and the ablation study limited to CIFAR-100 (iTW2); the AC found that the other concerns like impact of generation quality, computation analysis, implementation details were already addressed by the appendix of the initial submission.

The authors responded to these comments through the rebuttal and subsequent responses, which addressed many of the concerns successfully. The lack of comparisons with domain-invariance baselines has been fully resolved by the rebuttal, providing additional experiments and clear reasons why the proposed method outperformed the domain-invariance baselines. The concern with the assumption of a shared synthetic-real domain subspace seems to be well resolved by additional experiments and thorough discussions in the last response from the authors, as well as by results presented in the appendix, the reviewer did not mention the new additional results and discussions in their final justification though. The authors also conducted analysis on sensitivity to hyperparameters and reported results honestly; the proposed method was indeed sensitive to some extent, but the AC found that this cannot be a sole ground for rejection, particularly regarding that the method achieved state of the art across diverse benchmarks with the same hyperparameter setting.

The remaining concerns are incremental technical novelty and limited evaluation of robustness and generalization. The authors' response failed to fully convince reviewer iTW2's concern on the novelty as they still believed the proposed method is a combination of existing ideas and thus lacks groundbreaking innovation. The AC however found that the the authors' last response to this concern is solid and even without the technical novelty, the motivation and findings of this paper are valuable and worthy to be introduced to the incremental learning community. On the other hand, the concerns with limited evaluation of robustness and generalization were about lack of applications beyond simple classification tasks and relatively naive experimental settings. However, the AC considers that this issue cannot be a ground for rejection since the paper in the current form is already full of content and the reviewer who raised this concern was supportive of this paper in the end.

Putting these together, the AC found that the positive points and the rebuttal outweigh the remaining concerns, and thus recommend acceptance of this paper. The authors are strongly encouraged to reflect the valuable comments from the reviewers and to better establish clear connections between the main text and appendix.